# MCL1 modulates mTORC1 signaling to promote bioenergetics and tumorigenesis

Wentao Gui [1,2,3,4,5], Petr Paral [6,7], Bhavuk Dhamija[1,2], Eman Hagag[1,2], Martin Dusa[6], Jana Humajova[6], Pavla V. Francova [6], Jan Kucka [8], Jan Pankrac[6], Caroline Schütz[1,2], Vasileios Armenis[1,2], Filippo Ferrucci[1,2,3], Mario Schubert [9], Kaomei Guan [9], Franziska Baenke[10,11,12], Daniel E. Stange [10], Lorenz H. Lehmann[13,14,15], Wolfram Weckwerth [16,17], Peter Mirtschink[1], Sofia Traikov[1], Belmonte Giuseppe[18], Clelia Miracco[18], Martin Bornhäuser [2,11], Saverio Minucci[19], Ludek Sefc [6], Libor Macurek[4] & Mohamed Elgendy [1,2,3,4,11] ✉

Myeloid cell leukemia-1 (MCL1) is among the most overexpressed proteins in tumors. MCL1 contributes to tumorigenesis by antagonizing apoptosis. However, apoptosis-unrelated functions are emerging. Screening an array of signaling switches identifies mTORC1 to be modulated by MCL1 but not by the anti-apoptotic Bcl-2 or Bcl-xL. mTORC1 is a central metabolic regulator. MCL1 impacts metabolism via modulating the expression of hexokinase 2 (HK2) in an mTORC1-dependent manner, which ultimately contributes to the tumor-promoting effects of MCL1. MCL1 inhibitors suppress mTORC1 in tumor cells but are associated with cardiotoxicity due to mTORC1 inhibition in the heart. Dietary leucine supplementation rescues mTORC1 signaling in the hearts of humanized *Mcl-1* mice and greatly ameliorates the cardiotoxicity of MCL1 inhibitors. Taken together, here we describe tumor-promoting roles for MCL1 in regulating mTORC1 signaling and subsequently in bioenergetics, besides its role in antagonizing apoptosis, identifying MCL1 as a hinge of cell bioenergetics and survival.

Deregulated energy substrate metabolism and evading apoptosis are two hallmarks of cancer that need to be closely coordinated. Due to their sustained viability and fast proliferation, cancer cells reprogram their energy metabolism to meet the high metabolic demands needed to fuel survival and proliferation. Metabolic reprogramming has been linked to key oncogenic switches. For instance, mutational activation of K-Ras has been shown to lead to an increase in glycolytic activities[1–3].

Similarly, c-myc has been shown to drive the glutamine addiction of some tumors[4]. However, in addition to such metabolic shifts downstream of oncogenic switches, it is also plausible that common effectors that exert dual or multiple functions in cell survival, proliferation and metabolism may exist. Identification of such effectors may offer attractive drug targets.

MCL1 is a pro-survival member of the Bcl-2 family of proteins that stands out from the rest of the anti-apoptotic Bcl-2 family members by having unique features, including its short half-life and complex regulation[5–9]. MCL1 is regulated on multiple levels of transcription, post-transcription, translation, and post-translation[10–13]. Being a short-lived protein, MCL1 relies on constant de-novo protein translation to maintain its level and may thus be impacted—among many other short-lived proteins- by perturbation of mTORC1 signaling due to the global functions of mTORC1 signaling in protein translation[14–17]. Numerous studies have shown that inhibition of mTORC1 leads to suppression of protein translation, which ultimately impacts MCL1 levels leading to a decline in cell viability. Reducing the high level of MCL1 in tumors has been suggested as a

---

mechanism of action of several anti-cancer agents targeting mTOR signaling[18–21].

*MCL1* locus has been shown to be amplified in around 10% of all tumor entities[22] including melanoma[23,24] and acute myeloid leukemia (AML)[25,26]. MCL1 upregulation has been shown to contribute to survival, drug resistance and relapse of several types of tumors[27]. The contribution of MCL1 upregulation to tumorigenesis has been largely attributed to its role in allowing tumors to evade apoptosis. However, other apoptosis-unrelated roles of MCL1 are emerging. We and others have unraveled functions for MCL1 in autophagy[28–30], mitochondrial respiration[7–9] and coordinating the response to metabolic crisis[31], suggesting a tight link between MCL1 and cellular metabolism. Some reports have also linked MCL1 to nutrient (particularly glucose) sensing and metabolism. In those studies, MCL1 mediated survival and antagonized apoptosis in response to nutrient availability downstream of signaling pathways implicated in nutrient sensing and metabolism such as PI3K or GSK3β[32–34].

Targeting MCL1 is emerging as a potential therapeutic strategy[26,35,36] and therefore, deep understanding of MCL1 functions is particularly important and timely.

Here, we show that MCL1 regulates bioenergetics through modulation of the central metabolic regulator mTORC1 signaling pathway.

## Results

### MCL1 regulates mTORC1 signaling

Besides its established role in antagonizing apoptosis, the emerging apoptosis-unrelated functions of MCL1 challenge the classical view of MCL1 as being merely a downstream effector of molecular events and signaling cascades. This prompted us to examine whether MCL1 can impact some switches in cellular signaling cascades. Using two human phospho-kinase profiler kits, we examined the phosphorylation of a wide array of key signaling kinases shortly upon depleting MCL1 in CHL-1 cells, a melanoma cell line that expresses high levels of MCL1. While most of the examined proteins exhibited similar phosphorylation patterns in control and MCL1-depleted cells, Ribosomal protein S6 kinase beta-1 (S6K1) and Ribosomal protein S6, two downstream targets of mTORC1 signaling pathway, were among the few proteins exhibiting differences: In MCL1-depleted cells, phosphorylation of S6K1 and S6 was reduced compared to control, suggesting inhibition of mTORC1 signaling (Fig. 1A and Supplementary Fig. 1A, B).

To further explore this unexpected link between MCL1 level and mTORC1 signaling and assess whether it is a unique feature of MCL1 or a shared effect with other anti-apoptotic members of the Bcl-2 family of proteins, we examined by immunoblotting the phosphorylation of downstream targets of S6K1 and S6 in CHL-1 melanoma cells depleted of either MCL1, Bcl-2 or Bcl-xL. Our results show that depletion of MCL1, but not Bcl-2 or Bcl-xL, markedly inhibited mTORC1 signaling activity as indicated by the reduction of the phosphorylation of mTORC1 targets (Fig. 1B).

As mTORC1 signaling is a convergence point of many signaling cascades[37] and the mechanisms of mTORC1 regulation differ in different cells[38,39], we sought to examine the effect of MCL1 depletion on mTORC1 signaling in a number of melanoma cells with diverse mutational backgrounds. Besides CHL-1, depletion of MCL1 in SK-MEL30 (NRAS^mu/wt, TP53^mu/wt, CDKN2A^mu/mu), IGR-1 (BRAF^mu/mu, RAC1^mu/wt), IPC-298 (NRAS^mu/wt, TP53^mu/mu, CDKN2A^mu/mu) and MeWo (TP53^mu/mu, CDKN2A^mu/mu) cells led to marked inhibition of mTORC1 signaling in all models regardless of the mutational background (Fig. 1C–E and Supplementary Fig. 1C). Conversely, overexpression of MCL1 in melanocytes induced mTORC1 signaling (Supplementary Fig. 1D).

Furthermore, immunoblotting analysis of a panel of melanoma cell lines suggested a correlation between basal mTORC1 activity and the levels of MCL1-but not Bcl-2-in most of those cells (Supplementary

Fig. 1E), suggesting that MCL1 may play a role in regulating mTORC1 basal activity.

Besides melanoma, MCL1 plays crucial roles in promoting the survival and progression of acute myeloid leukemia (AML)[40,41] and colorectal cancer (CRC)[42]. Similar to the results obtained in melanoma, depletion of MCL1 in AML cells (MV4-11 and MOLM13) led to suppression of mTORC1 activity (Supplementary Fig. 1F). In HCT116 CRC cells transfected with doxycycline-inducible shRNA targeted against MCL1, gradual depletion of MCL1 upon addition of doxycycline led to concomitant inhibition of mTORC1 signaling (Supplementary Fig. 1G). These results establish a novel role for MCL1 in regulating mTORC1 signaling that seems to be a unique feature of MCL1 not shared with the closely related anti-apoptotic proteins Bcl-2 and Bcl-xL and suggest that it is a general phenomenon, not confined to a cell model or cancer entity. For further functional and mechanistic analyses, we focus on melanoma as a study model.

### MCL1 upregulation correlates positively with hyperactivation of mTORC1 signaling in melanoma patients

Both MCL1 and mTORC1 directly contribute to tumorigenesis. Elevated levels of MCL1 as well as mTORC1 hyperactivation are detected in several types of tumors[43–46]. We and others have shown that MCL1 upregulation[28] as well as mTORC1 activation[38,43,47] play crucial roles in melanoma progression. Given the link we observed between both oncogenic events, we performed immunohistochemical analysis on tissue samples derived from melanoma patients (Fig. 1F, G). This analysis established a significant correlation between the level of MCL1 and mTORC1 activity in melanoma: MCL1 was almost undetectable in nevi, and its low expression was associated with very low basal mTORC1 signaling activity, whereas elevated MCL1 levels in melanoma samples significantly correlated with the magnitude of activation of mTORC1 as indicated by the phosphorylation of its downstream target S6 protein (Spearman's rho = 0.67, *p* = 7.3e-07). In contrast, no significant correlation was found between the phosphorylation of ERK, a downstream target of the Ras/Raf pathway that is often deregulated in melanoma, and neither MCL1 level nor mTORC1 activity (pERK vs MCL1: *p* = 0.15, pERK vs pS6: *p* = 0.58). Furthermore, immunoblotting analysis of lysates prepared from either tumors or adjacent normal tissues from five melanoma patients showed a close correlation between the extent of MCL1 upregulation and the level of mTORC1 signaling in those tumors, further confirming the association between both oncogenic events (Fig. 1H).

### Modulation of mTORC1 by MCL1 is independent of apoptosis

Next, we sought to get insight into how MCL1 modulates mTORC1 signaling. Given the established role for MCL1 in the regulation of apoptosis, we initially examined whether mTORC1 inhibition upon MCL1 depletion was associated with or a result of the induction of apoptosis. We observed that the inhibition of mTORC1 signaling in MCL1-depleted cells is evident early after the transduction of cells with shRNA against MCL1 and before any detectable effect on cell viability. Immunoblotting analysis further confirmed that cells transduced with shRNA against MCL1 exhibited a significant inhibition of mTORC1 signaling 72 h post transduction but no sign of induction of apoptosis at this time as indicated by the absence of PARP cleavage, an established marker of apoptosis, in contrast to the PARP cleavage evident in positive control cells treated with apoptosis-inducer Actinomycin D (Supplementary Fig. 2A). Pan caspase inhibitor zVAD-fmk, at a concentration that effectively blocked apoptosis, failed to rescue mTORC1 inhibition in MCL1 depleted cells (Supplementary Fig. 2A).

Furthermore, depletion of MCL1 inhibited mTORC1 in BAX and BAK doubly-deficient mouse embryonic fibroblasts (Bax/Bak DKO MEFs), which fail to activate mitochondrial outer membrane permeabilization (Supplementary Fig. 2B).

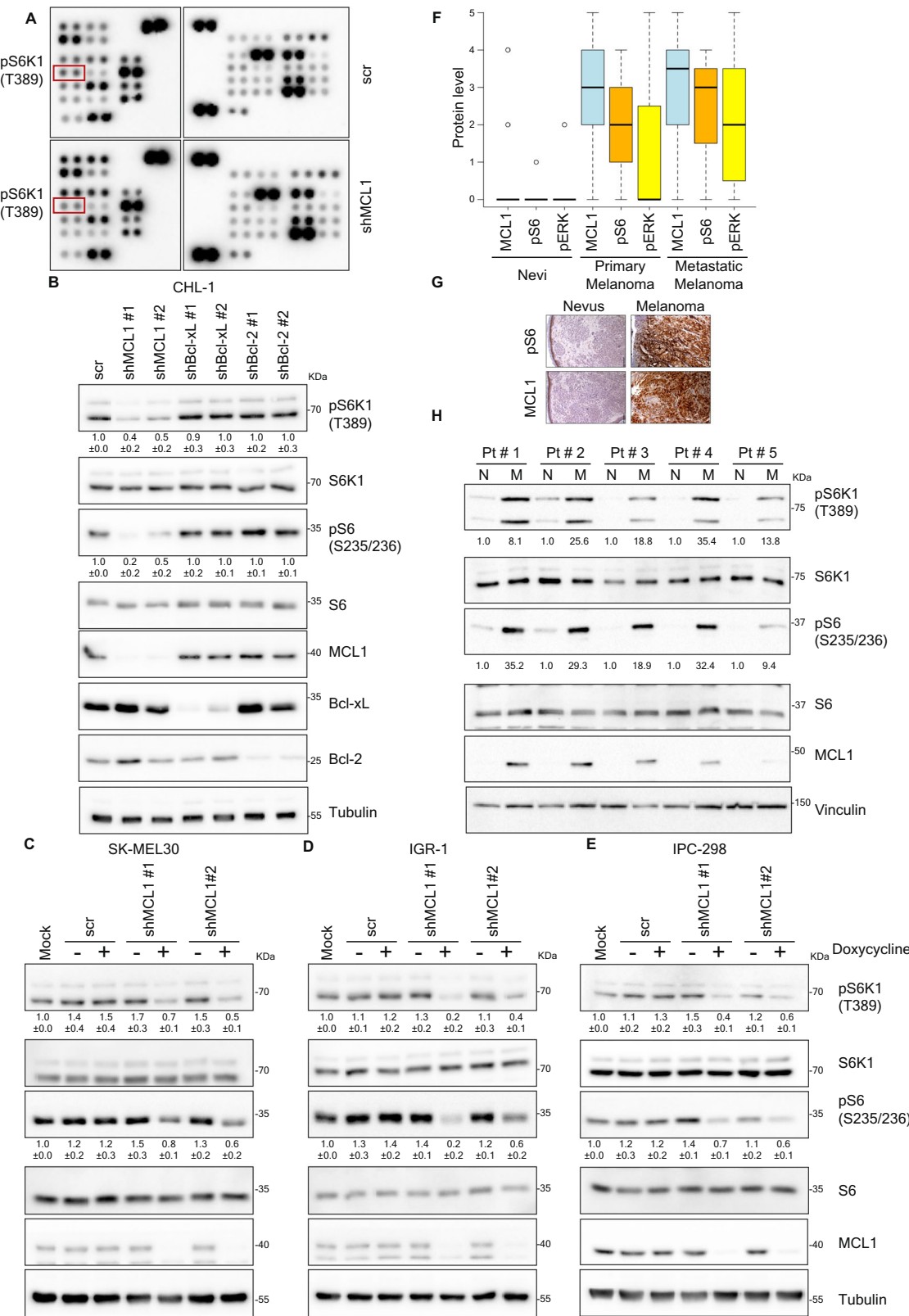

Finally, we exploited HEK-293T cells as a model of cells that do not rely on MCL1 for survival and whose viability is thus not affected upon MCL1 inhibition[48]. Similar to the results obtained in other cell models, depletion of MCL1 but not Bcl-2 or Bcl-xL led to suppression of mTORC1 in HEK-293T (Supplementary Fig. 2C). The magnitude of mTORC1 suppression closely correlated with the differential levels of

MCL1 knockdown achieved by four different shRNAs (Supplementary Fig. 2C).

Taken together, these results indicate that suppression of mTORC1 upon MCL1 depletion is independent of apoptosis and does not seem to correlate with the dependency on MCL1 for survival.

**Fig. 1 | MCL1 regulates mTORC1 signaling. A** Phospho-kinase array analysis of lysate derived from CHL-1 melanoma cells transduced either with scrambled shRNA or shRNA against MCL1 for 72 h, identifying modulation of mTORC1 target p70S6 Kinase by MCL1 depletion. **B** Immunoblotting analysis of lysate derived from CHL-1 cells transduced with the indicated shRNAs for 72 h. The samples derive from the same experiment but different gels for pS6K1, MCL1, Bcl-xL, Bcl-2, another for S6K, pS6 and another for S6 were processed in parallel. **C–E** Immunoblotting analysis of lysates derived from SK-MEL30 (**C**), IGR1 (**D**) and IPC298 (**E**) melanoma cells transduced with either scrambled shRNA or doxycycline-inducible shRNAs against MCL1 and treated with or without 500 ng/ml doxycycline for 72 h. The samples derived from the same experiment but different gels for pS6K1, pS6, MCL, and another for S6, S6K1 were processed in parallel. **F** Immunohistochemical analysis of MCL1, phospho S6 and phospho ERK levels in patient samples of nevi ($n = 14$ patients), primary melanoma ($n = 38$ patients) or metastatic melanoma ($n = 10$ patients). The boxplot displays the distribution of protein scores. The upper and lower whiskers indicate the maximal and minimal scores, respectively, excluding outliers, the boxes indicate the highest and lowest quartiles, the thick bars indicate the medians, and the circles indicate the outliers (more or less than 1.5 times the upper or lower quartile, respectively). **G** Representative images of immunohisto-chemical analysis in "F" (Original magnification x 200) Scale bar = 40 µm. **H** Immunoblotting analysis of lysates prepared from either tumor-adjacent normal (N) or malignant (M) tissues of five melanoma patients. The samples derived from the same experiment but different gels for pS6K1, pS6, MCL1, and another for S6, S6K1 were processed in parallel.

## MCL1 modulates signaling upstream of mTORC1 through regulating Sestrin 2 levels

mTORC1 plays crucial cellular functions and its activation is therefore tightly regulated by several mechanisms, including subcellular localization, complex assembly, protein-protein interaction and importantly through signal transduction from several upstream signaling cascades[49–53]. mTORC1 is primarily regulated by the Ras-like small GTPase Rheb[54]. Rheb must be in the GTP-bound state to activate mTORC1. GTP binding of Rheb is opposed by the GAP activity of the tuberous sclerosis complex (TSC), a heterodimer of the polypeptides Hamartin (TSC1) and Tuberin (TSC2)[54,55]. Several upstream signaling pathways converge at positive or negative regulation of the TSC complex and thereby inversely impact mTORC1 activity. We sought to examine whether MCL1 modulates signaling upstream of mTORC1. Immunoblotting analysis showed that in contrast to control, TSC2-depleted cells did not exhibit mTORC1 inhibition upon depletion of MCL1 (Fig. 2A). This indicates the requirement of TSC2 for mediating the effect of MCL1 on mTORC1 and suggests that MCL1 regulates signaling upstream of mTORC1. Among the switches that act upstream of TSC2, Sestrin 2 (SESN2) is an evolutionarily conserved regulator of mTORC1. Sestrin 2 activates TSC2 and therefore inhibits mTORC1 signaling[56]. Interestingly, we found that depletion of MCL1 but not Bcl-2 or Bcl-xL led to an increase in the levels of Sestrin 2 protein (Fig. 2B, C). Upregulation of Sestrin 2- but not Sestrin 3- by MCL1 depletion was observed in a panel of melanoma cell lines (Fig. 2B–D), arguing for the generalization of this effect in melanoma cells.

Given the role of Sestrin 2 in the regulation of mTORC1, we next aimed to test whether Sestrin 2 upregulation in MCL1-depleted cells may contribute to the observed mTORC1 inhibition. We examined the effect of simultaneous knockdown of both MCL1 and Sestrin 2 on mTORC1 signaling. Similar to the results obtained with TSC2 knockdown, Sestrin 2-depleted cells exhibited less mTORC1 inhibition upon depletion of MCL1 (Fig. 2E), establishing a role for Sestrin 2 upregulation upon MCL1 depletion in mediating the signaling between MCL1 and mTORC1 (Fig. 2F).

## MCL1 controls cellular bioenergetics in vitro and in vivo

mTORC1 plays a central role in regulating cellular metabolism and particularly bioenergetics through its ability to control glycolysis and oxidative phosphorylation (OXPHOS)[17,57–59]. With the unexpected effect of MCL1 on mTORC1 signaling, we next sought to explore potential functions of MCL1 in cellular bioenergetics.

Metabolic analysis using Seahorse metabolic analyzer showed that, compared to Bcl-2 and Bcl-xL, depletion of MCL1 resulted in a significant inhibition of both OXPHOS and glycolysis as assessed by the decline in oxygen consumption and extracellular acidification rates, respectively (Fig. 3A–E and Supplementary Fig. 3A–D). Consistently, MCL1 depletion also led to a decrease in lactate production and glucose consumption (Supplementary Fig. 3E, F), decline in mitochondria- and glycolysis- derived ATP production (Fig. 3F, G and Supplementary Fig. 3G), which altogether resulted in decrease in absolute levels of ATP

(Fig. 3H) that was not associated with increase in ADP or AMP levels (Supplementary Fig. 3H), indicating decrease in ATP production and energetic crisis in MCL1-depleted cells rather than increased ATP utilization.

As metabolism of cancer cells in vitro may differ from that of tumors in vivo, we aimed to assess the role of MCL1 in regulating tumor metabolism in vivo. Consistent with the in-vitro results, induction of MCL1 knockdown in tumor xenografts established from melanoma cells expressing doxycycline-inducible shRNA against MCL1 resulted in decrease in mTORC1 activity (Fig. 3I and Supplementary Fig. 4A), which was concomitant with decline in respiration (Fig. 3J and Supplementary Fig. 4B) and decline in ATP production rather than utilization (Fig. 3K and Supplementary Fig. 4C-E). Finally, we inoculated mice in both flanks with either control melanoma cells transduced with doxycycline-inducible scrambled shRNA or shRNA against MCL1. After establishment of tumors, mice were treated shortly with doxycycline to induce MCL1 knockdown. Mice were then subjected to 18-fluorodeoxyglucose positron emission computed tomography (18FDG PET), an imaging technique that assesses the uptake of radio-labeled 18FDG and is routinely used to monitor glucose metabolism in tumors. After euthanasia, 18FDG uptake in freshly isolated tumors was quantified using automated gamma counter. MCL1 depletion in tumors led to decline in 18FDG signal as compared to control tumors, suggestive of inhibition of glucose metabolism in MCL1-depleted tumors (Fig. 3L–N and Supplementary Fig. 4F, G). Of note, at the time of tumor isolation there was no significant difference in tumor size between both groups of tumors. Establishing the control and MCL1-depleted tumors on both flanks of mice controlled for the inter-mouse and inter-organ variabilities of 18FDG uptake (Supplementary Fig. 4H,I), as it allowed comparison of two tumors established in the same mouse. These results suggest that MCL1 plays a role in regulating tumor metabolism in vivo.

## MCL1 controls bioenergetics via modulating mTORC1 signaling

The inhibitory effect of MCL1 depletion on bioenergetics is independent of apoptosis as caspase inhibitor zVAD-fmk failed to rescue the decline in mitochondria- and glycolysis- derived ATP rates (Supplementary Fig. 4J–M) but was consistent with the observed inhibition of mTORC1, and with the central functions of mTORC1 in controlling energy metabolism[60–62]. However, as a decline in cellular bioenergetics may also lead to mTORC1 inhibition through activation of 5' adenosine monophosphate-activated protein kinase (AMPK), we aimed to distinguish whether the observed inactivation of mTORC1 is rather the cause or a manifestation of the inhibition of cellular bioenergetics upon MCL1 depletion. To this end, we examined the effect of rescuing mTORC1 suppression in MCL1 depleted cells by simultaneous depletion of TSC2 and MCL1 on cellular bioenergetics. Rescuing mTORC1 suppression reversed the inhibitory effect of MCL1 depletion on both mitochondria- and glycolysis- derived ATP rates (Fig. 3 O, P and Supplementary Fig. 4N, O), indicating that MCL1 acts to modulate mTORC1 signaling, which then in turn impacts bioenergetics.

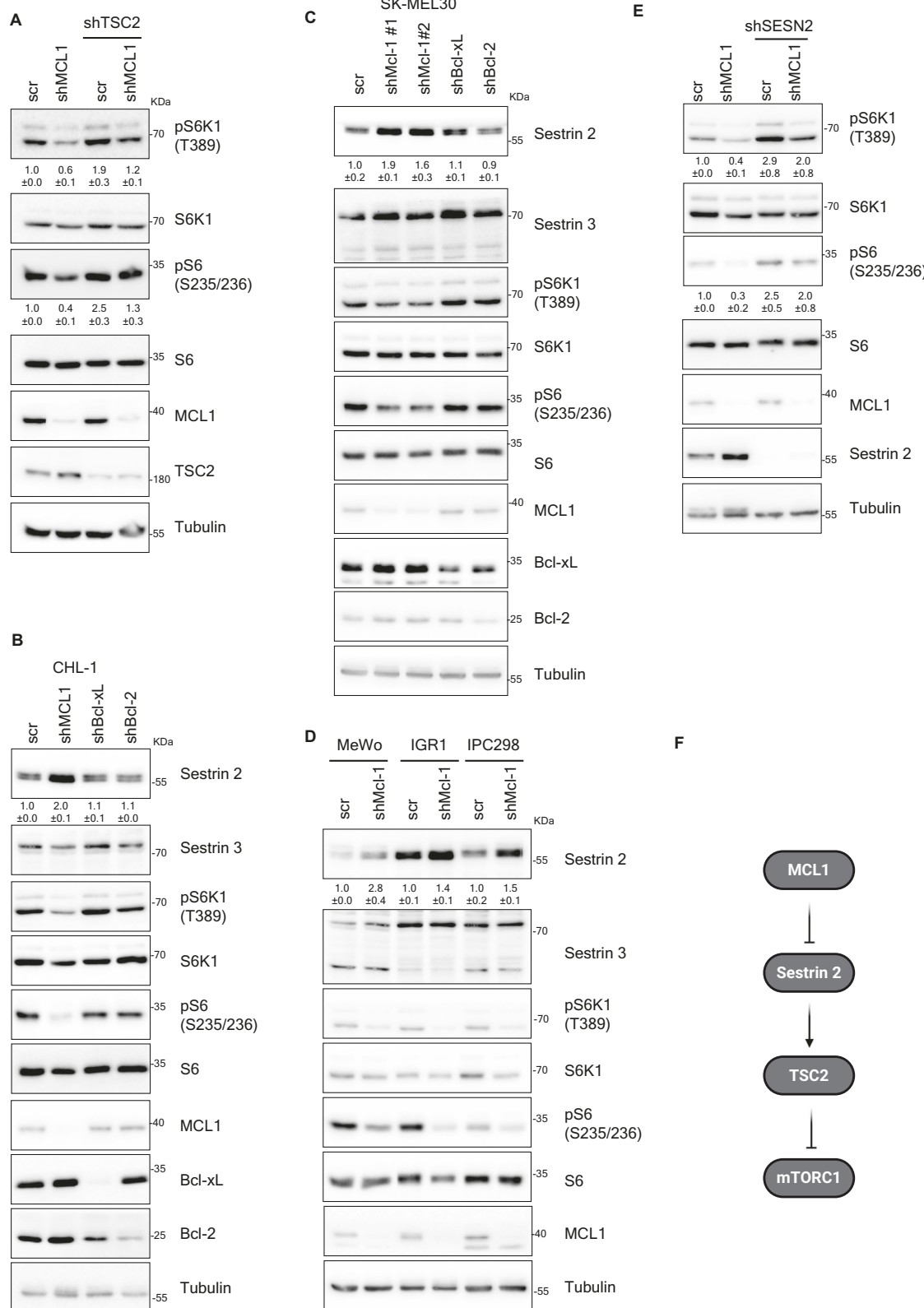

## MCL1-mTORC1 axis controls the levels of HK2

Next, we aimed to gain further mechanistic insight into how specifically the newly identified MCL1-mTORC1 axis impacts bioenergetics pathways. We assessed by real-time qPCR the transcriptional levels of an array of key metabolic regulators in cells depleted of either MCL1 alone or both MCL1 and TSC2 to identify the alterations induced by MCL1 in mTORC1-dependent manner. To further ensure that any rescue effect observed in the MCL1/TSC2 double knockdown cells is indeed due to the rescue of mTORC1 inactivation rather than other mTORC1-independent effects of TSC2 depletion, we also included an additional condition of cells doubly deficient of MCL1 and TSC2 and treated with mTOR inhibitor rapamycin. Our analysis showed that depletion of MCL1 led to decline in the level of hexokinase II (HK2), among the most significant alterations (Fig. 4A and Supplementary

**Fig. 2 | Modulation of mTORC1 by MCL1 is independent of apoptosis and is mediated by Sestrin 2-TSC2 signaling. A** Immunoblotting of lysate derived from CHL-1 cells transduced with the indicated shRNAs for 72 h showing that TSC2 knockdown rescues mTORC1 signaling activity in MCL1-depleted cells. The samples derived from the same experiment but different gels for pS6, MCL1, another for TSC2, pS6K1, S6 and another for S6K1 were processed in parallel.
**B–D** Immunoblotting of lysate derived from CHL-1 (**B**), SK-MEL30 (**C**), MeWo, IGR1 and IPC298 (**D**) cells transduced with the indicated shRNAs for 72 h showing an

increase in Sestrin 2 protein level in MCL1-depleted cells. The samples derived from the same experiment but different gels for Sestrin2, pS6, another for pS6K1, MCL1, Bcl-xL, another for Sestrin3, S6, Bcl-2 and another for S6K1 were processed in parallel. **E** Immunoblotting of lysate derived from CHL-1 cells transduced with the indicated shRNAs for 72 h showing that Sestrin 2 knockdown rescues mTORC1 signaling activity in MCL1-depleted cells. **F** Schematic representation of the model of mTORC1 modulation by MCL1 in melanoma cells. (Created in BioRender. Elgendy, M. (2025) https://BioRender.com/8hqhd6t).

---

Fig. 5A). HK2 is an isoform of hexokinase that catalyzes the rate-limiting and first obligatory step of glucose metabolism and has been shown to impact both glycolysis and mitochondrial respiration[63,64]. The downregulation of HK2 upon MCL1 depletion was mediated by mTORC1 inhibition as it was reversed upon rescuing mTORC1 inhibition by simultaneous MCL1/TSC2 knockdown and was restored again by inhibition of mTORC1 in MCL1/TSC2 doubly deficient cells by rapamycin treatment. Immunoblotting analysis of lysates from cells of the above mentioned conditions showed consistent modulation of HK2 on the protein levels and further showed that modulation of HK2 by MCL1 is not shared with Bcl-xL or Bcl-2 and was observed in a panel of melanoma cells (Fig. 4B, C and Supplementary Fig. 5B, C).

### MCL1-mTORC1 axis regulates bioenergetics via controlling HK2

With the crucial roles of HK2 in glucose metabolism and bioenergetics, we next aimed to assess whether the regulation of HK2 levels by MCL1-mTORC1 axis mediates -at least in part- the observed effect on bioenergetics. Rescuing HK2 downregulation in MCL1-depleted cells by ectopic overexpression of HK2 -but not two other metabolic regulators PKM2 and ALDOA- partially reversed the decline in mitochondria- as well as glycolysis- derived ATP rate (Fig. 5A, B and Supplementary Fig. 6A, B), indicating that HK2 downregulation upon MCL1 depletion plays a role in mediating the inhibitory effect on bioenergetics. Of note, while partially rescuing the bioenergetic crisis, overexpression of HK2 did not alter the suppression of mTORC1 in MCL1-depleted cells (Fig. 5C), further confirming that HK2 acts downstream of the MCL1-mTORC1 axis to modulate bioenergetics.

### Control of HK2 expression contributes to the tumor–promoting functions of MCL1

The established role of MCL1 in antagonizing apoptosis is undoubtedly crucial for the tumor- promoting functions of MCL1. Besides evasion of apoptosis, dysregulation of cellular bioenergetics is another emerging hallmark of tumors[65]. With the unraveled functions of MCL1 in regulating bioenergetics through modulation of HK2 levels in mTORC1-dependent manner, we aimed to test whether this could be another apoptosis-unrelated mechanism by which MCL1 contributes to tumorigenesis. In melanoma cells, MCL1 depletion led to a decline in cell viability on longer term. HK2 overexpression in MCL1-depleted cells, which as shown earlier rescues the bioenergetic inhibition, partially rescued the decline in viability of MCL1-depleted cells (Fig. 5D and Supplementary Fig. 6C). We further assessed the relevance of these in-vitro results in in-vivo xenograft model. NSG mice were inoculated with melanoma cells transduced with control or doxycycline-inducible shRNA against MCL1 either alone or with HK2 overexpressing vector. Upon establishment of tumors, mice were treated with doxycycline to induce the depletion of MCL1. Consistent with the in vitro results, tumor growth was impeded by MCL1 depletion and HK2 overexpression partially restored the growth of MCL1-depleted tumors (Fig. 5E–G), suggesting a role for modulation of HK2-mediated bioenergetics in the tumor-promoting effects of MCL1.

Finally, analysis of database of melanoma patients showed a significant correlation between the expression levels of HK2 and MCL1 ($p = 2.3e-06$, rho = 0.216) –but not BCL-2 ($p = 1.98e-01$, rho = 0.059) or BCL2L1 ($p = 4.57e-01$, rho = 0.034) (Fig. 5 H-J). This correlation seems to

be specific to HK2 as MCL1 expression does not correlate positively with neither PKM2 nor ALDOA (Supplementary Fig. 6D-I).

### Pharmacological inhibitors of MCL1 modulate mTORC1 signaling

Our mechanistic analysis established a role for Sestrin 2 in mediating the regulation of mTORC1 by MCL1. Interestingly, immunoprecipitation analysis showed that MCL1 binds Sestrin 2 and that the BH3 binding pocket of MCL1 is required for this binding as overexpression of wild-type MCL1-but not a mutant construct lacking the BH3 binding pocket co- immunoprecipitated with Sestrin 2 and induced mTORC1 signaling (Supplementary Fig. 7A, B). BH3-mimetics that can bind and inhibit the BH3 binding pocket of MCL1 are under clinical development for therapeutic intervention in cancer[25,36,66–68]. We aimed to test whether pharmacological inhibitors of MCL1 reproduce the effect of MCL1 depletion on mTORC1 signaling. Treatment of melanoma cells with two specific[69,70] and potent MCL1 inhibitors ABBV-467[71] and UMI-77[35,72,73], resulted in inhibition of mTORC1 in dose- and time-dependent manner (Fig. 6A–F). Concurrently, treatment with ABT-737, which targets Bcl-2 and Bcl-xL but not MCL1, did not evoke similar effect (Fig. 6C).

### mTORC1 inhibition contributes to cardiotoxicity of MCL1 inhibitors

The suppressive effect of MCL1 inhibition on mTORC1 in cancer cells described here could be a desirable−although unintended or unexpected- effect, given the tumor-promoting function of mTORC1[74–76]. However, MCL1 is also expressed in other healthy tissues to varying extents and inhibition of MCL1 may therefore impact mTORC1 signaling and metabolism in metabolically- active organs, which ultimately may contribute to the side effects of MCL1 inhibitors.

mTOR plays crucial roles as a sensor of nutrients. Several amino acids signal to mTOR, among which, leucine is an important regulator of mTORC1. Leucine has been suggested to modulate mTORC1 through several mechanisms, including negating mTORC1 suppressors Sestrin 1 and 2, ultimately resulting in mTORC1 activation[77,78]. Interestingly, leucine supplementation rescued mTORC1 inhibition in MCL1- depleted 293 T cells in a dose-dependent manner (Supplementary Fig. 7C). We aimed to analyze the effect of MCL1 inhibition on different tissues and importantly, test whether rescuing mTORC1 signaling by leucine supplementation may potentially reverse any observed effect. First, doses of ABBV-467 that effectively limited growth of melanoma xenografts were associated with dose-dependent inhibition of mTORC1 in tumors (Supplementary Fig. 8A and B). Next, we aimed to analyze the contribution of mTORC1 inhibition in other organs to the reported toxicity of MCL1 inhibitors. A major challenge in the preclinical analysis of the toxicity of MCL1 inhibitors is their several fold higher affinity towards human MCL1 compared to murine MCL1[79], which greatly limited the translation of preclinical data to human studies. To address this crucial limitation, we made use of humanized Mcl-1 (huMcl-1) mice, in which MCL-1 was replaced with its human homolog, allowing precise predictions of efficacy and tolerability for clinical translation[79]. huMcl-1 mice were treated with MCL1 inhibitor ABBV-467, alone or in combination with leucine supplementation to

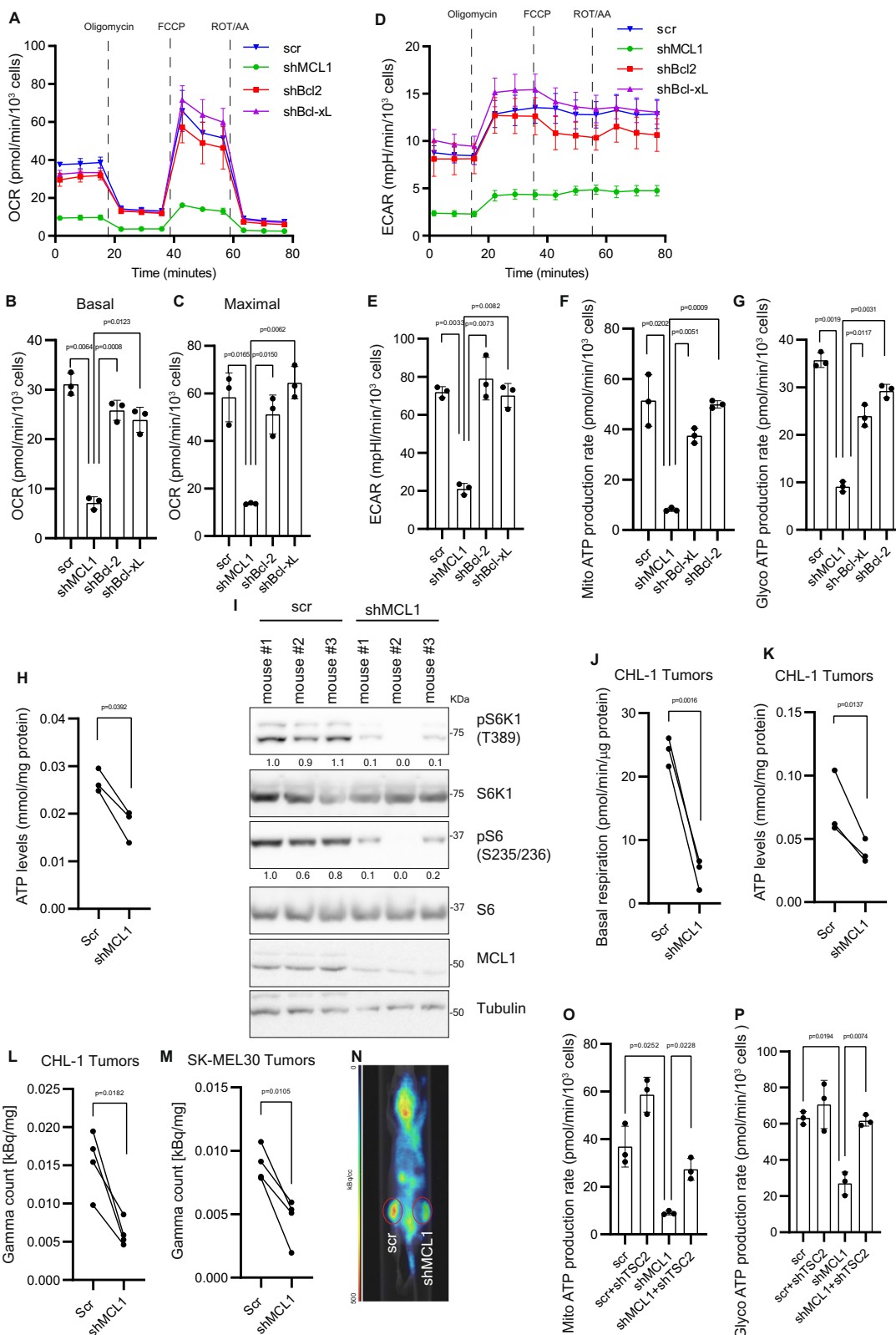

assess the effect of rescuing mTORC1. Additionally, a group of mice receiving ABBV-467 and leucine supplementation were further treated with rapamycin to confirm the specific association of mTOR signaling to any effect that may be exerted by leucine. Consistent with what has been observed in patients[71], *huMcl-1* mice treated with ABBV-467 showed signs of cardiotoxicity as indicated by dose-dependent increase in plasma cardiac-specific Troponin T levels (Fig. 7A) and

decrease in cardiac output (Supplementary Fig. 8C and Supplementary Table 1). Higher dose of ABBV-467 was associated with a decrease in white blood cells count and an increase in red blood cells and platelets counts, reported as pharmacodynamic markers[36]. (Supplementary Fig. 8 D-F). However, observed cardiotoxicity seemed to be specific as no significant differences in kidney or liver functions were observed (Supplementary Fig. 8G and Supplementary Table 2).

**Fig. 3 | MCL1 controls cellular bioenergetics via mTORC1. A**–**C** Oxygen consumption rate (OCR) (**A**), basal (**B**) and maximal (**C**) respiration measured by Seahorse XF Mito Stress Test of CHL-1 cells transduced with the indicated shRNAs for 72 h. **D, E** Extracellular acidification rate (ECAR) of CHL1 cells measured as in (**A**). (*n* = 3 biologically independent samples). Mitochondria- (**F**) and glycolysis- derived (**G**) ATP production rate measured by Seahorse XF Real-Time ATP Rate Assay of CHL-1 cells transduced with the indicated shRNAs for 72 h. (*n* = 3 biologically independent samples). **H** ATP levels measured using LC-MS/MS in CHL-1 cells transduced with either scrambled shRNA or shRNA against MCL1. Lines connect values from biologically independent repeats (*n* = 3 biologically independent samples). **I** Immunoblotting analysis of lysates prepared from subcutaneous xenografts established in both flanks of mice with CHL-1 cells transduced with either scrambled shRNA or doxycycline-inducible shRNA against MCL1. The samples derived from the same experiment but different gels for pS6 and pS6K, another for S6K1 and S6 and another for MCL1 were processed in parallel. **J** Basal mitochondrial respiration of tumors derived from CHL-1 cells as in (**I**) (*n* = 3 mice per group). Tumors were isolated, immediately dissociated using GentleMACS

tissue dissociator, plated in poly-D-lysine coated Seahorse plates and measured by Seahorse XF. **K** ATP levels in tumors derived from CHL-1 cells as in (**I**) measured using LC-MS/MS and. Lines connect values from the two tumors established on both flanks of the same mouse. *n* = 3 mice per group). **L, M** Qualification of gamma radiation count normalized to administrated ¹⁸FDG and dry weight [kBq/mg] of control (scr) and MCL1-depleted (shMCL1) tumors established as in (**I**) from CHL-1 (**L**) or SK-MEL30 (**M**) cells followed by ¹⁸FDG-PET assay and finally isolation of tumors and measurement of gamma radiation using automated gamma counter. Lines connect values from the two tumors established on both flanks of the same mouse. (*n* = 4 mice per group). **N** Representative image of coronal plane of mice bearing control or MCL1-depleted tumors on both flanks (circled) established as in (**I**) followed by ¹⁸FDG-PET assay. Mitochondria- (**O**) and glycolysis- derived (**P**) ATP production rate measured by Seahorse XF Real-Time ATP Rate Assay of CHL-1 cells transduced as indicated. (*n* = 3 biologically independent samples). Statistics are derived from 3 or more biological replicates. Data is presented as mean +/- SD and significance is determined by paired two-tailed t-test.

Remarkably, leucine supplementation significantly ameliorated the cardiotoxicity of ABBV-467 as indicated by higher cardiac output and lower cardiac Troponin T levels in *huMcl-1* mice receiving leucine supplementation. This effect was mediated by modulation of mTOR signaling as rapamycin treatment almost completely abolished the cardio-protective effect of leucine (Fig. 7 A, B). Furthermore, ABBV-467 decreased mitochondrial respiration in the hearts of *huMcl-1* mice, which was reversed by leucine supplementation in mTOR-dependent manner (Fig. 7C).

Immunoblotting analysis of lysates prepared from hearts of *huMcl-1* mice from different groups further confirmed that treatment with ABBV-467 was associated with inhibition of mTORC1 and consistent with its cardio-protective effect, leucine supplementation rescued mTORC1 signaling and rapamycin abolished this rescue (Fig. 7D).

Finally, similar effects were observed with another inhibitor of MCL1 UMI-77. Effective doses of UMI-77 impeded tumor growth and were associated with mTORC1 inhibition in tumors but similarly triggered cardiotoxicity that was reversed by leucine supplementation, an effect that was abolished by rapamycin (Supplementary Fig. 9 A-D and Supplementary Table 3). Immunoblotting of lysates from hearts further confirmed the association with modulation of mTORC1 activity (Supplementary Fig. 9E). These data indicate an association between mTORC1 inhibition induced by MCL1 inhibitors and cardiotoxicity, and importantly indicate that leucine supplementation could potentially serve cardio-protective functions.

## Discussion

Deregulated energetics is an emerging hallmark of cancer that is associated with the cancer cell property of evading cell death. On the one hand, cancer cells need to coordinate their bioenergetic program to guarantee the production of energy needed to fuel survival and proliferation. On the other hand, bioenergetics can be impacted by pro-survival and pro-proliferative stimuli downstream of central oncogenic switches. Identification of common regulators that exert dual or multiple functions in cell survival and metabolism can offer synergistic therapeutic targets.

MCL1 is one of the most overexpressed proteins in several tumor entities. Its tumor-promoting role has so far been largely attributed to its function as an anti-apoptotic member of the Bcl-2 family. Here we describe an unexpected role for MCL1 in controlling mTORC1 signaling and subsequently in regulating cellular bioenergetics, independently of its role in antagonizing apoptosis. Taken together, this may position MCL1 as an important link between bioenergetics and cell survival by acting as a connecting circuit. The unique features of MCL1 of short half-life and tight regulation make it an ideal switch to initiate or terminate signaling cascades depending on the fluctuation of nutrient levels and the need to coordinate cell survival accordingly.

Emerging reports implicated novel functions for MCL1 in the regulation of mitochondrial dynamics[7–9]. Given the tight links between mitochondrial dynamics and OXPHOS, it could be extrapolated that MCL1 may impact OXPHOS. However, the finding that MCL1 also impacts glycolysis in vitro and in vivo that we show here, cannot be directly explained by its mitochondrial functions and rather suggests that MCL1 acts as a central regulator of cellular bioenergetics, which we show to be through its capacity to modulate the central metabolism circuit mTORC1. Indeed, both OXPHOS and glycolysis are regulated by different downstream effectors of mTORC1[17,57–59]. Importantly, several lines of evidence indicate that the effect of MCL1 on mTORC1 signaling is independent of apoptosis: MCL1 depletion results in mTORC1 inhibition in HEK 293 T cells which do not rely on MCL1 for survival and do not show cell death upon MCL1 knockdown. The same effect was also observed in Bax/Bax DKO cells. Even in cells that rely on MCL1 for survival, such as melanoma, depletion of MCL1 or treatment with lower doses of MCL1 inhibitors lead to drastic inhibition of mTORC1 much earlier before the detection of any cellular or molecular signs of cell death and is not modulated by blocking caspase activation. Further, no modulation of mTORC1 signaling is observed in cells depleted of Bcl-2 or Bcl-xL (two other modulators of apoptosis) or in cells treated with ABT-737, which targets Bcl-2 and Bcl-xL but not MCL1. Moreover, mTORC1 modulation by MCL1 can be rescued by perturbation of TSC2 or Sestrin 2. Finally, ectopic overexpression of MCL1 induces mTORC1 activation in melanocytes and the levels of MCL1 overexpression correlate tightly with mTORC1 signaling activity in melanoma patient samples.

Besides its upstream functions in controlling bioenergetics, mTORC1 also acts as an important sensor of nutrient and energy levels and can be regulated by a drop in energetics through AMPK activation. However, several observations made in this study establish that the modulation of mTORC1 by MCL1 is a result of direct crosstalk rather than just a manifestation of a change of energetic status. This establishes a role for MCL1 in modulation of mTORC1 and HK2 upstream of its impact on metabolism. Firstly, the inhibition of both glycolysis and OXPHOS can be rescued by impeding the inhibition of mTORC1 through depletion of mTORC1 suppressor TSC2. Secondly, upon MCL1 knockdown, cells overexpressing HK2 show less drop in bioenergetics but comparable mTORC1 inhibition, positioning HK2 and bioenergetics downstream of MCL1-mTORC1 axis. Mechanistic insights on the identified link between MCL1 and mTORC1 implicates a role for Sestrin 2 downstream of MCL1 and upstream of mTORC1 and a role for HK2 modulation downstream of MCL1-mTORC1 axis and upstream of bioenergetics pathways.

Further mechanistic analysis indicates the physical binding between Sestrin 2 and MCL1 plays a role in mediating the effect on mTORC1. It will be interesting in future studies to map this interaction,

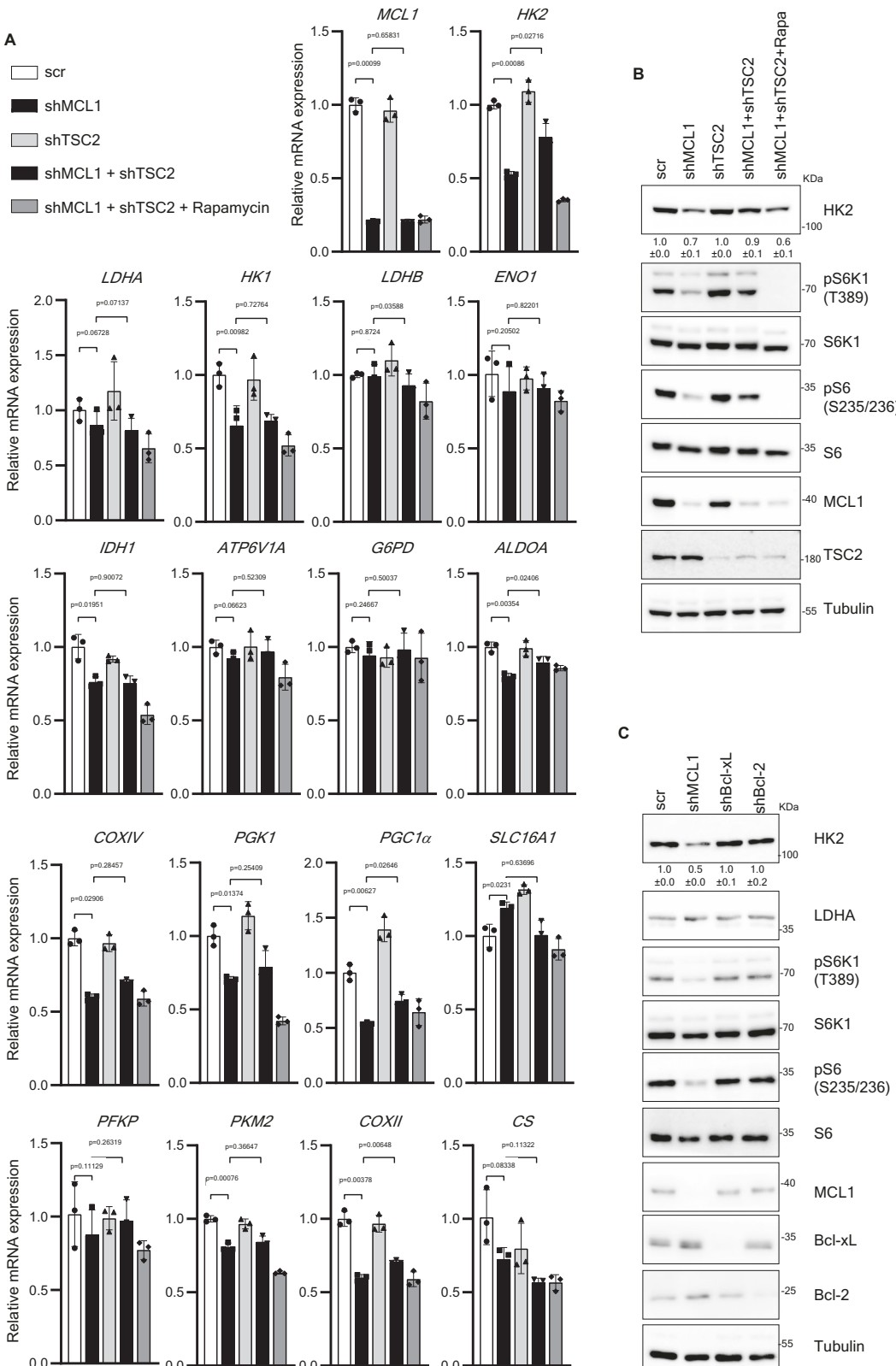

**Fig. 4 | MCL1 modulates HK2 in mTORC1-dependent manner. A** Relative mRNA expression levels (normalized to beta-actin) of an array of metabolic regulators assessed by real-time qPCR in CHL-1 cells transduced with the indicated shRNAs and treated with or without Rapamycin (50 nM) for 24 h (*n* = 3 biologically independent samples). Data is presented as mean +/- SEM and significance is determined by paired two-tailed t-test. **B** Immunoblotting analysis of total cell lysates from CHL-1 cells treated as in (A) showing modulation of HK2 protein level by MCL1-mTORC1 axis. The samples derived from the same experiment but different gels for HK2, S6K1, MCL1, another for TSC2, pS6K1, S6 and another for pS6 were processed in parallel (**C**) Immunoblotting analysis of lysates of CHL-1 cells transduced with either scrambled shRNA or shRNA against MCL1, Bcl-2 or Bcl-xL showing specific modulation of HK2 levels by MCL1. The samples derived from the same experiment but different gels for HK2, pS6K1, MCL1, Bcl-xL, another for pS6, LDHA, another for S6K1, Bcl-2, and another for S6 were processed in parallel.

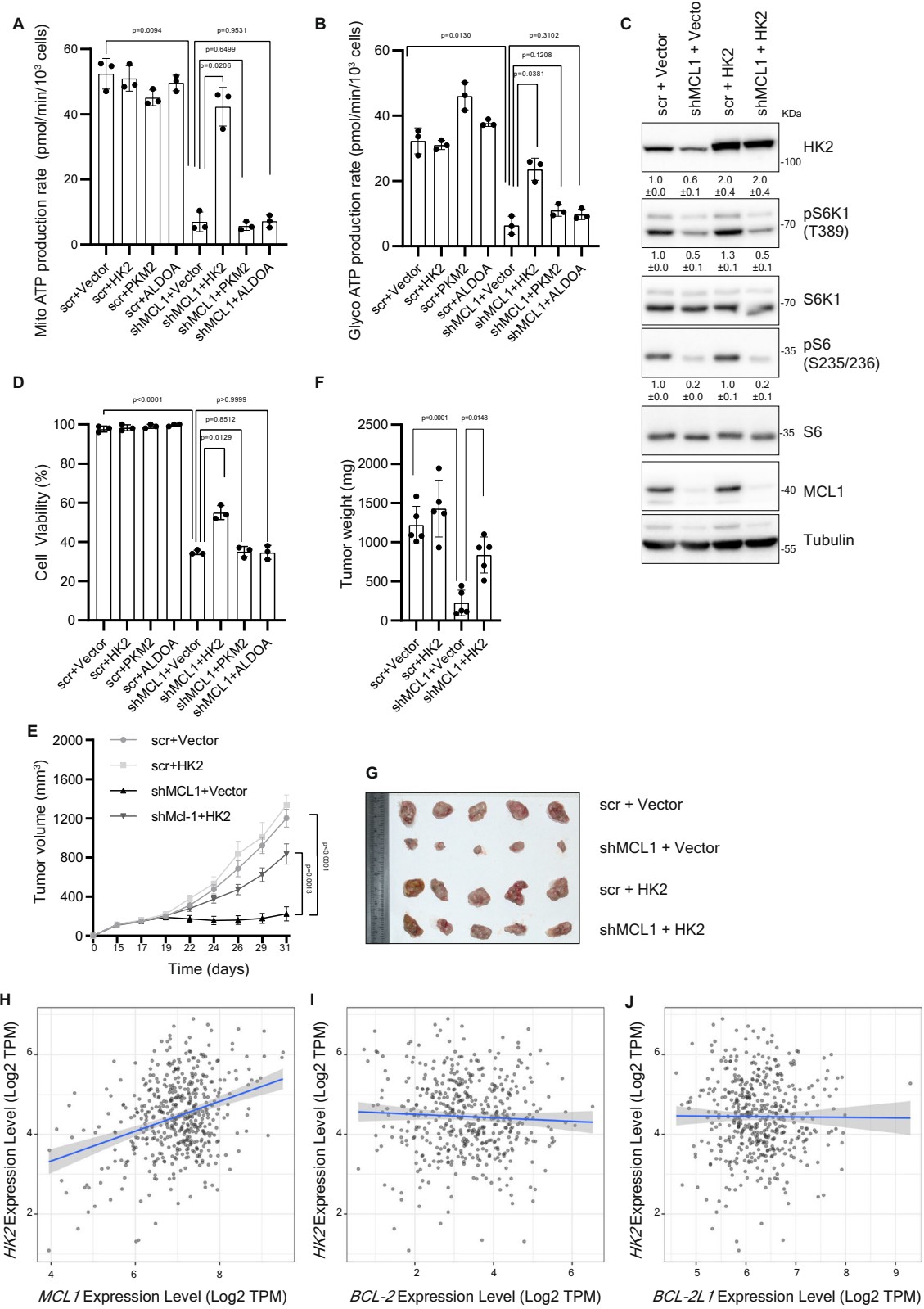

distinguish whether it is a direct interaction or through a common interacting protein. Additionally, it will also be crucial to assess the functional implications on other signaling cascades downstream of Sestrin 2, as well as on the interplay with apoptosis.

Interestingly, physical interactions have been shown to mediate the functions of MCL1 in several processes besides apoptosis. Physical interaction between MCL1 and specific long-chain acyl-

coenzyme A (CoA) synthetases of the ACSL family ACSL1 has recently been shown to mediate the novel function of MCL1 in long-chain fatty acid β-oxidation (FAO)[80]. A previous report has shown that interaction between MCL1 and very long-chain acyl-CoA dehydrogenase (VLCAD), a key enzyme of the mitochondrial fatty acid β-oxidation pathway, induces the enzymatic activity of the latter which ultimately mediates the role of MCL1 in the dynamic regulation of

**Fig. 5 | Modulation of HK2 contributes to the role of MCL1 in promoting bioenergetics and tumorigenesis.** Mitochondria- (**A**) and glycolysis- derived (**B**) ATP production rate measured by Seahorse XF Real-Time ATP Rate Assay of CHL-1 cells overexpressing glycolysis regulators HK2, PKM2, ALDOA or vector and transduced with either scrambled shRNA or shRNA against MCL1 for 72 h. (*n* = 3 biologically independent samples). Data is presented as mean +/- SD and significance is determined by paired two-tailed t-test. **C** Immunoblotting analysis of lysates derived from CHL-1 cells overexpressing HK2 or vector and transduced with either scrambled shRNA or shRNA against MCL1 for 72 h. The samples derived from the same experiment but different gels for HK2, pS6K1, MCL1, Tubulin, another for S6K1, pS6 and another for S6 were processed in parallel. **D** Percentage of cell

viability of CHL-1 cells transduced as in (**A**) after 96 h in culture. (*n* = 3 biologically independent samples). Growth rate (**E**), weight (**F**) and images (**G**) of subcutaneous xenografts established in NSG mice from CHL-1 cells overexpressing HK2 or vector and transduced with either scrambled shRNA or doxycycline-inducible shRNA against MCL1. After establishment of xenografts, mice were kept on 1 mg/ml doxycycline supplemented in the drinking water to induce MCL1 shRNA. (*n* = 5 mice per group). Data is presented as mean +/- SEM and significance is determined by unpaired two-tailed t-test. Correlation between the mRNA levels of HK2 and MCL1 (**H**), BCL-2 (**I**) and BCL2L1 (**J**) in The Cancer Genome Atlas (TCGA TCGA-SKCM (*n* = 471 patients) analyzed using TIMER2.0 (http://timer.cistrome.org/). Spearman's rho value was used to evaluate the degree of their correlation.

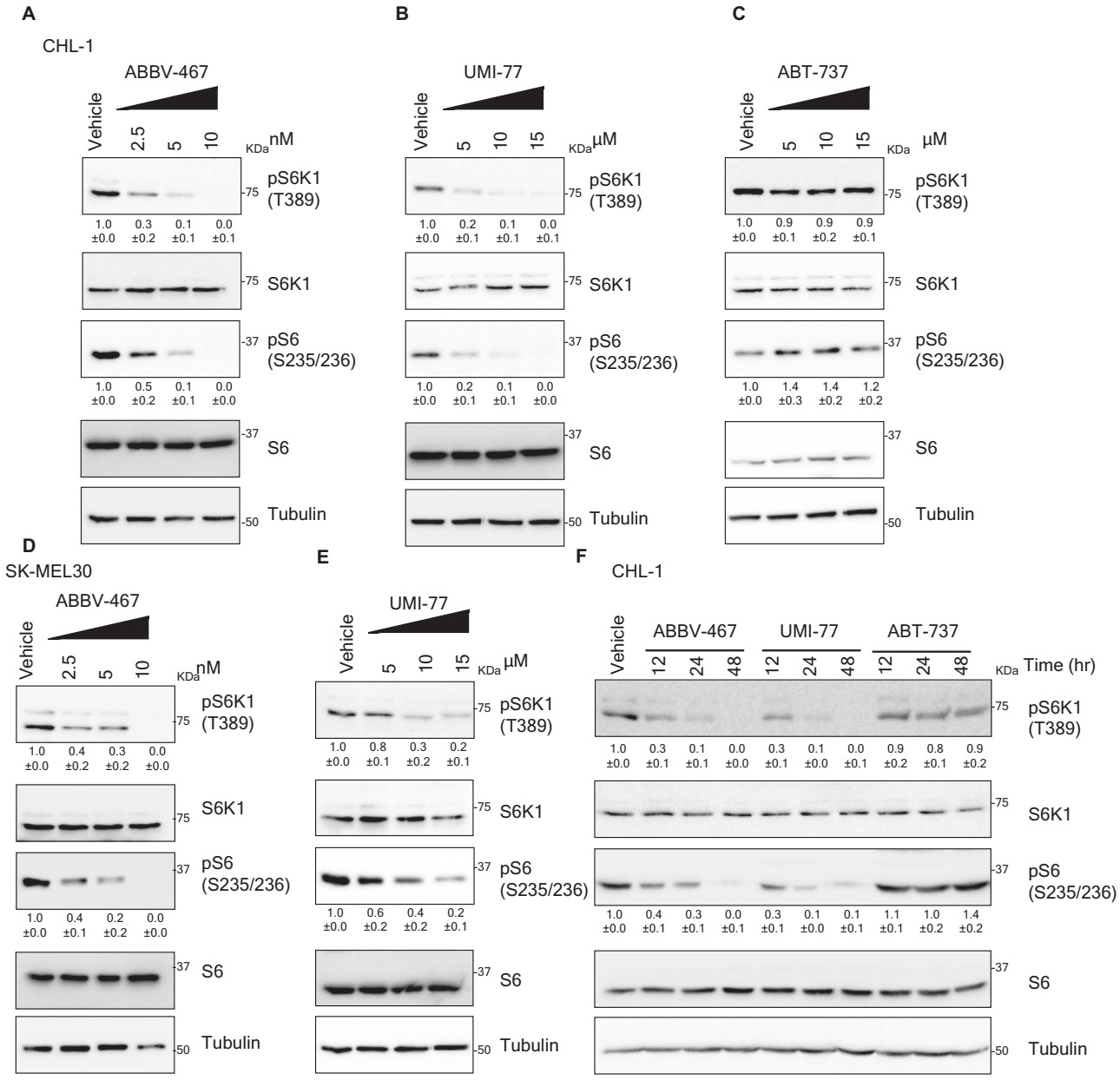

**Fig. 6 | MCL1 pharmacological inhibitors suppress mTORC1.** Immunoblotting analysis of lysates derived from CHL-1 cells (**A**–**C**) or SK-MEL30 cells (**D**, **E**) treated with the indicated concentrations of MCL1 inhibitors ABBV-467, UMI-77 or as control ABT-737 for 48 h. The samples derived from the same experiment but different gels for pS6K1, pS6 and another for S6K1, S6 were processed in parallel.

**F** Immunoblotting analysis of lysates derived from CHL-1 cells treated ABBV-467 (10 nM), UMI-77 (10 μM) or ABT-737 (10 μM) for the indicated time points. The samples derived from the same experiment but different gels for pS6 and pS6K, another for S6K1 and another for S6 were processed in parallel.

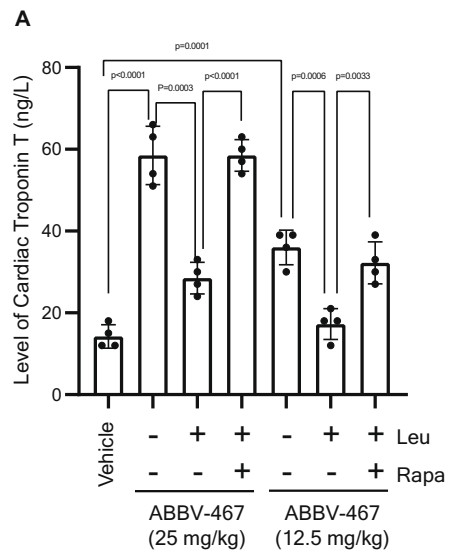

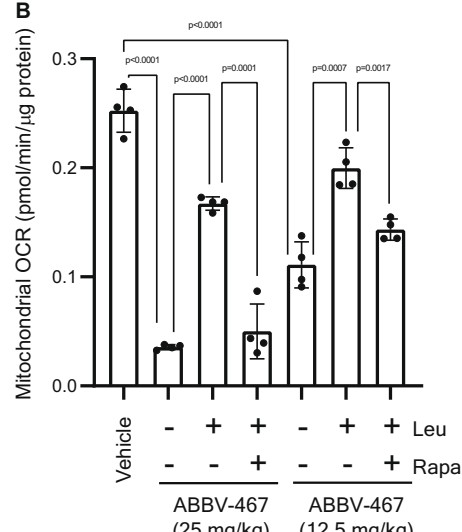

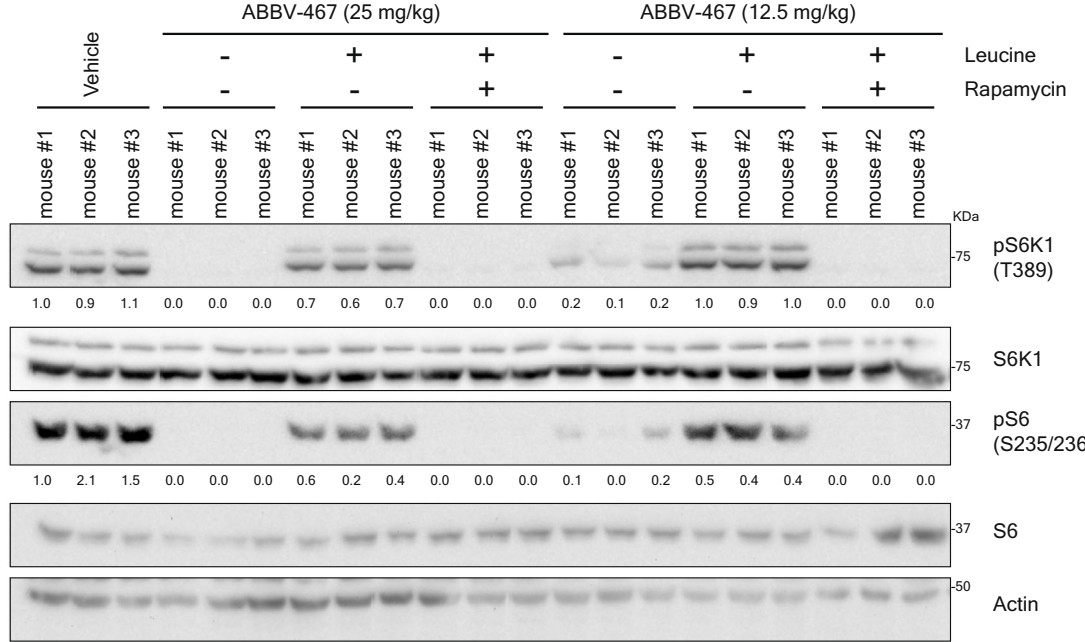

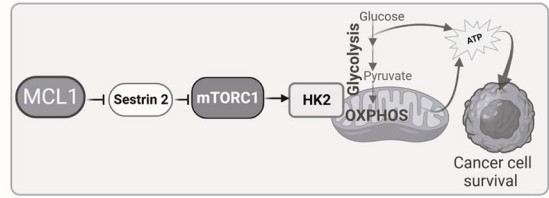

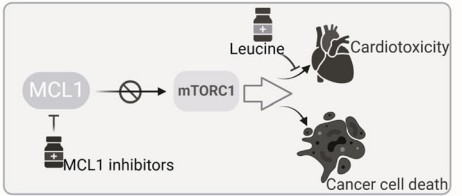

lipid metabolism. Additionally, interaction with MCL1 has been shown to enhance the stability of DRP-1 and OPA1, two GTPases responsible for remodeling of the mitochondrial network, which mediates the effect of MCL1 on mitochondrial dynamics[81]. Deeper understanding of these functions is an important future direction, especially in the light of ongoing efforts of targeting MCL1 for therapeutic intervention in cancer.

MCL1 inhibitors are currently under clinical development. However, cardiotoxicity has been reported to be a major obstacle in clinical testing[82,83]. MCL1 is expressed in cardiomyocytes and MCL1 deletion has been shown to result in cardiomyopathy[84,85]. Interestingly, inhibition of apoptosis by ablation of Bax and Bak failed to reverse the mitochondrial abnormalities in MCL1-deficient hearts, ruling out a significant link with apoptosis. Further investigation implicated the

**Fig. 7 | mTORC1 inactivation mediates cardiotoxicity of MCL1 inhibition and can be ameliorated by leucine supplementation. A** Serum levels of cardiac-specific Troponin T of humanized Mcl-1 mice (*n* = 4 mice per group) treated for three weeks with Vehicle or ABBV-467 (25 mg/kg or 12.5 mg/kg administered by I.V. injection on a Q7D × 3 schedule) alone or in the indicated combinations with supplementation of Leucine (150 mmol/L in the drinking water) and treatment with Rapamycin (2 mg/kg I.P. three times a week). (*n* = 4 mice per group).
**B** Mitochondrial OCR of hearts isolated from (**A**) and immediately sliced, placed in Agilent Seahorse XF24 Islet Capture Microplate and measured by Seahorse XF24

metabolic Analyzer. (*n* = 4 mice per group). **C** Immunoblotting of lysate derived from the hearts of mice in A. The samples derived from the same experiment but different gels for pS6 and pS6K, another for S6K1 and another for S6 were processed in parallel. **D** Schematic representation (generated by biorender.com) of the model of MCL-1-mediated regulation of mTORC1 and bioenergetics and the effect of MCL1 inhibitors. Statistics are derived from 4 biological replicates. Data is presented as mean +/- SD and significance is determined by unpaired two-tailed t-test. Created in BioRender. Elgendy, M. (2025) https://BioRender.com/37rx81t and Elgendy, M. (2025) https://BioRender.com/rydxylv.

direct functions of MCL1 on mitochondrial dynamics[9]. Our results identify a role for mTORC1 suppression in the cardiotoxicity induced by MCL1 inhibitors.

mTORC1 is essential for the preservation of cardiac structure, growth, and vascular integrity in both prenatal and postnatal stages[86]. mTORC1 signaling plays key roles in cardiac energy metabolism through regulating fatty acid metabolism, glucose uptake, and glycolysis, and mitochondrial function[87]. Mice with deficient myocardial mTORC1 activity by targeted ablation of mTORC1 essential component raptor suffered from dilated cardiomyopathy and deteriorated cardiac functions 4 weeks after raptor ablation[88]. The attenuated mTORC1 activity critically affected cardiac protein and energy metabolism, mitochondrial content and structure, apoptosis, and autophagy, and rapidly led to cardiac failure[88]. In this model, after transverse aortic constriction, raptor deletion resulted in significantly decreased mitochondrial DNA, swollen mitochondria with irregular cristae, consistent with the role of mTORC1 in the control of mitochondrial regulators YY1 and PGC1α[89].

Furthermore, cardiac specific, kinase-dead mTOR transgenic mice showed significantly reduced cardiac function[90]. mTOR signaling has been shown to play a role in a mouse model of ischemia/reperfusion injury and mTOR inhibition by rapamycin increased injury[91]. Intriguingly, in challenge to these studies, mTORC1 activation in the heart during chronic stress has also been shown to have multiple maladaptive effects, such as the promotion of pathological hypertrophy, misfolded protein accumulation, and energy stress[86]. mTORC1 reduces cardiac remodeling and heart failure in response to pressure overload and chronic myocardial infarction. A study identified a cardioprotective role for rapamycin treatment in a mouse model of type 2 diabetes[92]. Dosing and timing of rapamycin may account for this discrepancy[88]. mTORC1 activation via inducible cardiac-specific TSC2 knockdown results in hypertrophy without contractile dysfunction[93,94]. However, while mTORC1 evidently becomes activated upon TSC2 ablation, other mTOR-independent effects of TSC2 inactivation have been reported[95,96]. Moreover, drugs that reduce mTOR activity are only partially successful in the treatment of TSC, suggesting that mTOR-independent pathways play a role in disease development[95]. Interestingly, comparing cardiac specific kinase-dead (kd) mTOR and constitutively active (ca) mTOR transgenic mice showed that the cardiac hypertrophic growth in response to physiological and pathological stimuli was not different in mTORkd and mTORca transgenic mice when compared with that of non-transgenic littermates, suggesting that the mTOR-mediated signaling pathway is not essential to cardiac hypertrophic growth[90]. The final contribution of mTORC1 to the physiology and pathology of unstressed and stressed hearts is therefore likely to be context-dependent. Differences in the specific models used, experimental conditions, treatment duration and dosage may all play crucial roles.

Modulation of mTOR signaling has also been linked with the cardiotoxicity of anti-cancer therapy. mTOR was identified in a screening of targeted therapy as a key kinase regulating cardiomyocyte survival[97]. Acute doxorubicin-induced cardiotoxicity (DIC) has been shown to be tightly associated with inhibition of the mTOR pathway[98]. In DIC context, activation of mTOR has been shown to mediate the cardio-protective roles of the transplanted embryonic

stem cells, the conditioned medium[99] as well as ribonucleotide reductase M2 subunit[100]. Conversely, inhibition of mTOR by alternate day fasting (ADF) in mice mediated the aggravation of ADF on DIC[101]. Taken altogether, numerous studies established a crucial role for mTOR signaling in the physiology and pathology of the heart. With the newly-discovered effects of MCL1 inhibitors in modulating mTORC1 signaling we describe here, our findings show a potential mechanism of the increasingly-reported cardiotoxicity associated with the use of some MCL1 inhibitors in clinical testing. In addition, the novel function of MCL1 in FAO has recently been shown to contribute to the cardiotoxicity triggered by MCL1 inhibitors[80], suggesting that additional cooperating mechanisms may also exist. While this has been shown to be mediated by the distribution of MCL1-ACSL1 interaction, the established functions of mTORC1 in FAO suggest a link[80]. It, however, remains to be established whether the two mechanisms are interconnected or act independently in parallel.

Importantly, we designed a dietary approach based on high leucine supplementation that rescued mTORC1 signaling and consequently ameliorated the cardiotoxicity induced by MCL1 inhibitors. Our data suggest that leucine supplementation could be a feasible cardio-protective approach that merits future clinical testing.

## Methods

We confirm that our research complies with all relevant ethical regulations. The animal experiments were performed in accordance with national and international guidelines for laboratory animal care, approved by the Laboratory Animal Care and Use Committee of the First Faculty of Medicine, Charles University in Prague, and the Ministry of Education, Youth and Sports and of the Czech Republic (MSMT-46307/2020-3) and the Ethic Committee of the Czech Academy of Science (AVCR 5020/2022 SOV).

### Cell culture

Melanoma and HEK-293T cells and MEFs were grown in Dulbecco's modified Eagle's medium (DMEM) supplemented with 10% FBS, 2mM L-glutamine and Penicillin/ streptomycin. Melanocytes were cultured in Ham's F10 medium supplemented with 10 ng/ml tetra-decanoylphorbol 13-acetate (TPA), 0.1 mM 3-isobutyl-methyl-xanthine (IBMX), 1% vol/vol Ultroser G, 2mM L-glutamine. AML cell lines were cultured in RPMI medium supplemented with 10% fetal bovine serum and 2mM L-glutamine unless otherwise indicated. All the cell lines were maintained at 37 °C and 5% CO2 and were regularly tested for mycoplasma contamination. Cells were treated with inhibitors ABT-737 (Selleck Chem), ABBV-467 (MedChemExpress), UMI-77 (Selleck Chem), Rapamycin (Sigma-Aldrich) and L-Leucine (Sigma-Aldrich).

### Lentivirus production and infection

shRNA lentiviral particles were produced and transduced following The RNAi Consortium (TRC) protocols. Briefly, HEK-293T packaging cells growing in 100 mm dishes were transfected at 60– 70% of confluence with a mix of 4.5 µg psPAX2 vector (packaging vector), 1.5 µg pMD2.G vector (envelope vector) and 6 µg hairpin-pLKO.1 vector. PEI-Max was used as transfection reagent according to the manufacturer's instructions. Cell culture medium was harvested three times for intervals of 24 h. Medium containing lentiviral particles was harvested

and filtered and used to transduce cells in the presence of 4 µg/ml polybrene (Sigma-Aldrich). The medium was replaced by fresh medium after 24 h. The oligonucleotide sequences used for shRNA were cloned into the pLKO.1-TRC cloning vector and the doxycycline-inducible Tet-pLKO-Puro vector according to manufacturer's instructions.

Sequences of the shRNAs were:

pLKO.1 scrambled shRNA Target sequence: CAACAAGATGAAGAGCACCAA

pLKO.1 human MCL1 shRNA #1 Target sequence: GCCTAGTTTATCACCAATAAT

pLKO.1 human MCL1 shRNA #2 Target sequence: GCAGGATTGTGACTCTCATTT

pLKO.1 human MCL1 shRNA #3 Target sequence: CTGATAACTATGCAGGTTTAA

pLKO.1 human MCL1 shRNA #4 Target sequence: GCTGTGTTAAACCTCAGAGTT

pLKO.1 human Bcl-xL shRNA #1 Target sequence: GTGGAACTCTATGGGAACAAT

pLKO.1 human Bcl-xL shRNA #2 Target sequence: GCTCACTCTTCAGTCGGAAAT

pLKO.1 human Bcl-2 shRNA #1 Target sequence: GTGATGAAGTACATCCATTAT

pLKO.1 human Bcl-2 shRNA #2 Target sequence: TGGATGACTGAGTACCTGAAC

pLKO.1 human Sestrin 2 shRNA Target sequence: GCGGAACCTCAAGGTCTATAT

pLKO.1 human TSC2 shRNA Target sequence: CACTGGCCTTGGACGGTATTG

## Constructs

-pLJC6-HK2-3xFLAG was a gift from Jason Cantor (Addgene plasmid # 163451

http://n2t.net/addgene:163451; RRID:Addgene_163451)[102].

-pEGFP-C1-PKM2 was a gift from Axel Ullrich (Addgene plasmid # 64698

http://n2t.net/addgene:64698; RRID:Addgene_64698)[103].

-pBoBi-hALDOA was a gift from Sheng-cai Lin (Addgene plasmid # 210806;

http://n2t.net/addgene:210806; RRID:Addgene_210806)[104].

-pLKO.1 - TRC cloning vector was a gift from David Root (Addgene plasmid # 10878;

http://n2t.net/addgene:10878; RRID:Addgene_10878)[105].

-Tet-pLKO-puro was a gift from Dmitri Wiederschain (Addgene plasmid # 21915;

http://n2t.net/addgene:21915; RRID:Addgene_21915)[106].

## Quantitative RT-PCR

Total RNA was isolated from cells using NucleoSpin RNA Plus kit (Macherey-Nagel). cDNA Synthesis was performed using with the iScript reverse transcription kit (Bio-Rad). qPCR was carried out using the SsoFast Eva Green Supermix (Bio-Rad), a CFX384 real-time System C1000 Thermal Cycler (Bio-Rad) and the Bio-Rad CFX Manager 3.1 software. Relative gene expression was calculated using the ΔΔCt method.

Primer sequence of qRT–PCR primers used in this study are as follows:

MCL1 5′GGACATCAAAAACGAAGACG3′, 5′GCAGCTTTCTTGGTTTATGG3´

LDHA 5′TTGACCTACGTGGCTTGGAAG3′, 5′GGTAACGGAATCGGGCTGAAT3′

LDHB 5′TGGCGTGTGCTATCAGCATT3′, 5′GCTTATCTTCCAAAACATCCACAAG3′

PGC1α 5′GAGGGAAAGTGAGCGATTAG 3′,5′ GTGAGGCTGATGTGTACTG3′

PKM2 5′AAGGGTGTGAACCTTCCTGG3′, 5′GCTCGACCCCAAACTTCAGA3′

PGK1 5′AAACTTTTGGACAGGACCACAGA3′,5′GCATCAGCCACTGGAACCA3′,

SLC16A1 5′CACTTAAAATGCCACCAGCA3′, 5′AGAGAAGCCGATGGAAATGA3′

ENO1 5′GCCTCCTGCTCAAAGTCAAC3′, 5′AACGATGAGACACCATGACG3′

IDH1 5′CACCAAATGGCACCATACGAA3′, 5′CCCCATAAGCATGACGACCTAT3′

HK1 5′CACATGGAGTCCGAGGTTTATG3′, 5′CGTGAATCCCACAGGTAACTTC3′

HK2 5′TGCAGCGCATCAAGGAGAACAAAG3′,5′ACGGTCTTATGTAGACGCTTGGC3′

COXIV 5′CGTTATCATGTGGCAGAAGC3′, 5′ATGGGGTTCACCTTCATGTC3′,

COXII 5′AGAGGGTAGAGCCGTTTCTTAG3′, 5′GCGTGTGAAAGGGTTCGAG3′

ATP6V1A 5′GAGATCCTGTACTTCGCACTGG3′, 5′GGGGATGTAGATGCTTTGGGT3′

PFKP 5′CGCCTACCTCAACGTGGTG3′, 5′ACCTCCAGAACGAAGGTCCTC3′

ALDOA 5′ATGCCCTACCAATATCCAGCA3′, 5′GCTCCCAGTGGACTCATCTG3′

G6PD 5′ACCGCATCGACCACTACCT3′, 5′TGGGGCCGAAGATCCTGTT3′

CS 5′TGCTTCCTCCACGAATTTGAAA3′, 5′CCACCATACATCATGTCCACAG3′

## Phosphokinase profiler array

Relative levels of phosphorylation of kinase phosphorylation sites and related total proteins were compared in lysates derived from CHL-1 melanoma cells transduced with scrambled shRNA or shRNA against MCL1 for 72 h using the human phospho-kinase array kits (ARY003C, R&D Systems and Human Phosphorylation Pathway Profiling Array C55, RayBiotech) according to the manufacturer's instructions. Briefly $3 \times 10^6$ cells were rinsed with PBS and collected in 300 µl of the provided lysis buffer. Lysate was cleared by centrifugation at $13,000 \times g$ for 5 minutes at 4 °C. Total protein concentration was quantified using Pierce BCA Protein Assay Kit (Thermo Scientific). Equal amount of protein of each sample was incubated overnight with the nitrocellulose membranes with the printed capture antibodies followed by washing steps and incubation with the corresponding biotinylated antibody cocktail. Signals were revealed using Streptavidin-HRP and Chemi reagent mix and imaged using the Fusion FX Imaging system (PeqLab Biotechnologie).

## Immunoblotting

Cells and tissue samples were lysed on ice in lysis buffer (50 mM Tris-HCl (pH7.4), 5 mM EDTA, 150 mM NaCl, 1% NP-40 supplemented with protease inhibitor cocktail (1:50), PMSF 200 mM (1:100), and sodium orthovanadate solution 100 mM (1:100)). Lysates were cleared by centrifugation at $13,000 \times g$ for 10 min at 4 °C. Protein concentration was measured using Pierce BCA Protein Assay Kit (Thermo Scientific). Protein samples were mixed with 6× Reducing Laemmli buffer, denatured at 95 °C for 5 min and loaded on 10% SDS-polyacrylamide gel. Proteins were then transferred on nitrocellulose membrane (Amersham Protran,GE Healthcare Lifescience). Membranes were blocked in 5% skimmed milk or bovine serum albumin (BSA) in TBS-T (0.1% Tween-20 in 1× Tris-buffered saline) for 1 h at room temperature and probed with target specific primary antibody overnight at 4 °C. After washing, membranes were incubated for 1 h at room temperature with the corresponding secondary antibodies (diluted 1:3000). The signal was detected using the enhanced chemiluminescence (ECL) western blot substrate (Thermo Scientific) or the Ultra-Sensitive HRP Substrate

(Takara) and imaged using the Amersham ImageQuan 800 (Cytiva Life Sciences) Imaging system and Azure western blot imaging system (Azure Biosystems). The following antibodies were used for immuno-blotting: anti-Mcl1 (Santa Cruz, sc-819), anti-Bcl2 (Santa Cruz, sc-7382), anti-Bcl-xL (CST, catalog no. 2764), anti-total p70S6kinase (CST, catalog no. 2708), anti-phospho- p70S6kinase (Thr389) (CST, catalog no. 9234), anti-S6 Ribosomal Protein (CST, catalog no. 2217), anti-phospho-S6 Ribosomal Protein (Ser235/236) (CST, catalog no. 2211), anti-Hexokinase II (CST, catalog no. 2867), anti-Tubulin (Santa Cruz, sc-271314), anti-β-Actin (CST, catalog no. 4970) and anti-Sestrin 2 (Proteintech, 10795-1-AP). Quantification of the intensity of the bands is expressed under blots. For pS6K and pS6, the values represent the relative band density to the total S6K or S6 proteins, respectively. For other proteins, the values represent the relative band density to the loading control.

## Immunohistochemistry
Immunohistochemical staining was performed on 3-mm-thick sections of the most representative tumor paraffin block, using the streptavidin-biotin method. Sections were dewaxed, pretreated in a bath (Lab Vision PT Module) (Bio-Optica, Milan, Italy), with WCAP citrate buffer, pH 6.0 (Bio- Optica), for 40 min at 98.5 °C, and incubated with 3% $H_2O_2$ in TBS to inhibit endogenous peroxidase. The primary antibodies were then applied at working dilution of 1:50. To improve the immuno-reactivity of antigens, we used the UltraVision LP Large Volume Detection System HRPor AP Polymer (ReadyTo-Use) (Bio-Optica), with either diaminobenzidine (Dako, Milan, Italy) for 8 min or new fuchsin for 7 min (Bio-Optica), as chromogens, respectively. All sections were then counterstained with Meyer's haematoxylin. For each antibody, negative controls were obtained by replacing the primary antibody with a non-immune serum at the same concentration.

Immunostaining was semi-quantitatively evaluated based on both the percentage of positive tumor cells (0, ≤ 10%; 1, > 10% and ≤ 50%; 2, > 50%) and the intensity of stain (0, absent; 1, weak; 2, moderate; 3, strong).

## Cell viability assay
Cell viability was assessed using 0.004 % Resazurin sodium salt (Sigma Aldrich) diluted in the culture medium. The cell viability was expressed as relative values compared to the control sample, which was defined as 100%.

## Animal experiments
Animals used in different in vivo experiments were housed in accordance with the approved guidelines (in individually ventilated cages with the sterilized bedding, 12:12 h light–dark cycle; at 22 ± 1 °C, and 60 ± 5% humidity), food and water provided ad libitum. Animals were regularly observed during the whole experiment for changes in their behavior and health status. The mice (C57Bl/6, hu*Mcl-1*, NSG) were bred in the SPF breeding facility in the Center for Experimental Biomodels, First Faculty of Medicine, Charles University (CEB). The experiments were performed in accordance with national and international guidelines for laboratory animal care, approved by the Laboratory Animal Care and Use Committee of the First Faculty of Medicine, Charles University in Prague, and the Ministry of Education, Youth and Sports and of the Czech Republic (MSMT-46307/2020-3) and the Ethic Committee of the Czech Academy of Science (AVCR 5020/2022 SOV). The maximal tumour size of 1.5 cm³ permitted by the ethics committee was not exceeded.

## Xenografts
NOD.Cg-Prkdcscid Il2 rgtm1 Wjl/SzJ mice (NSG mice) were originally purchased from The Jackson Laboratory (Bar Harbor, Maine, USA). NSG mice received single subcutaneous flank injections of 1*10⁶ melanoma cells transduced with the different constructs and suspended in 200 ml PBS. Tumor growth was monitored by bidimensional measurements using a caliper. Tumor volume was calculated as (length × width × width)/2. Animal experimentation was approved by the local ethics committee of IMG (reference number 39-2022-P). bitors.

Male humanized *Mcl-1* mice (8-week old) were treated with once weekly intravenous (I.V.) injection of ABBV-467 (25 or 12.5 mg/kg) formulated in a mixture of 5% DMSO, 10% cremophor EL, and 85% D5W. Male C57BL/6 mice (8-week old) were treated with tri-weekly intra-peritoneal (I.P.) injections UMI-77 (50 or 25 mg/kg) formulated in a mixture of 10% DMSO, 30% PEG300, and 60%ddH20. Leucine (150 mmol/L) was freshly supplemented in the drinking water. A group of mice treated with ABBV-467/UMI-77 and receiving leucine supplementation were treated with tri-weekly I.P. injections of rapamycin (2 mg/kg). Orbital sinus blood sample collection was done by inserting a capillary into the medial canthus of the eye (30° angle to the nose). The activity of cardiac Troponin T, alanine-amino transferase (ALAT), as well as creatinine in mouse serum were measured in a Cobas 8000 Analyzer (Roche Diagnostics, Germany) using Roche reagents following the manufacturer's instructions.

## ¹⁸FDG-PET imaging and gamma counting
Female Nod skid gamma (NSG) mice (8 weeks old) received single subcutaneous flank injections with 1 ×10⁶ CHL-1 or SK-Mel30 cells (in 200 ml saline) transduced with either scrambled shRNA or shRNA against MCL1 on both flanks. Once tumors were established, mice were given drinking water containing 1 mg/ml doxycycline and 1% sucrose for induction of MCL1 shRNA expression in vivo for five additional days. Mice were fasted overnight (14–18 h), weighed and heated prior to intravenous injection of ¹⁸FDG through the tail vein (activity approx. 6 MBq per mouse). Animals were placed into heated induction chamber under anaesthesia (1.5% isoflurane) for 40 min uptake. The 10 min PET imaging (Albira, Bruker, Germany) under anaesthesia (1.5% isoflurane) was performed running PET measurement (10 min, PET single, offset 35 mm). Blood glucose was measured using Accu-chek (Roche). Following euthanasia, tumors and internal organs were immediately isolated and the ¹⁸FDG activity was measured using a 2480 Wizard2® Automatic Gamma Counter (PerkinElmer, USA). Values were corrected for the half-life decay and normalized for injected ¹⁸FDG. PET image analysis and co-registration were carried out using PMOD analysis software (PMOD Technologies LLC; Switzerland). The mice were bred in the Center for Experimental Biomodels, First Faculty of Medicine, Charles University (CEB). The experiments were performed in accordance with national and international guidelines for laboratory animal care, approved by the Laboratory Animal Care and Use Committee of the First Faculty of Medicine, Charles University in Prague, and the Ministry of Education, Youth and Sports and of the Czech Republic (MSMT-46307/2020-3).

## Metabolic profiling with Seahorse metabolic analyzer
Oxygen consumption rate (OCR) and Extracellular acidification rate (ECAR) were measured simultaneously in melanoma cells using the XF96 Extracellular flux analyzer (Agilent Technologies). Cells were plated in XF96 cell culture microplate (Agilent) in the regular cell culture media. One hour before the assay, media were replaced by the XF Base Medium supplemented with glucose (10 mM), pyruvate (1 mM), and glutamine (2 mM). For Mito Stress Test, three basal measurements of OCR and ECAR were obtained before a sequential injection of oligomycin (2 μM final concentration), FCCP (1 μM final concentration) and a mixture of Rotenone +Antimycin A (0,5 μM final concentration). For real-time ATP rate assay oligomycin (1.5 μM final concentration) and a mixture of Rotenone and Antimycin A (0.5 μM final concentration) were used. All values were normalized to the cell numbers in each well and data were analyzed using wave software

(Agilent). For the measurement of respiration of tumors, tumors cells were immediately dissociated using GentleMACS (GentleMACS, USA) and were plated in poly-D-Lysin coated Seahorse plates and analyzed by Seahorse Metabolic Analyzer. Measurement of Mitochondrial OCR in the heart tissues of mice was performed according to the protocol described in (Kluza et al., 2021)[107]. Briefly, transversal thin slices of freshly-isolated hearts were prepared. Size-matched thin heart slices were placed in Agilent Seahorse XF24 Islet Capture Microplate and measured by Seahorse XF24 metabolic Analyzer. A mix of rotenone and antimycin A (at a final concentration of 10 μM for both inhibitors) was injected in order to inhibit mitochondrial respiration. Mitochondrial OCR was calculated from basal OCR (obtained without inhibitor) subtracted to OCR level after injection (non-mitochondrial respiration).

### Lactate production assay
Lactate level was measured in the cell culture media after 24 h of incubation. Briefly, samples were subjected to trichloroacetic acid (TCA) precipitation of proteins and centrifuged at $13,000 \times g$ for 5 min at 4 °C. Supernatant was diluted in assay buffer and transferred to transparent 96 well plate. Cell-free supernatants were incubated in lactate assay buffer (NAD+ substrate (0.6 mg/ml) plus 17 U/ml Lactate Dehydrogenase in Tris-Glycine-Hydrazine, pH 9.0) for one hour at 37 °C. Absorption at 340 nm was read to detect NADH accumulation[108,109], by the Synergy HT microplate reader. Data were normalized to the cell number.

### Glucose consumption assay
Glucose levels were measured in the cell culture media after 24 h of incubation. Briefly, glucose in presence of glucose oxidase enzyme forms gluconic acid and H2O2, this further reacts with a substrate (0-dianisdine) to give a colored product which can be read spectrometrically at 540 nm. Supernatant was diluted in 0.1 M Phosphate Buffer (pH 6) and transferred to transparent 96 well plate. Diluted cell free supernatant (10 μl) was incubated with 40 μl assay buffer and 50 μl reaction mix to make total reaction volume of 100 μl. Reaction mix comprises of o-dianisidine substrate (1:100), Horse radish peroxidase (1:500) and glucose oxidase (1:50). Plate was incubated for 30 minutes at 37 °C and reaction was stopped using conc. HCl to obtain pink colored product which is read at 540 nm. Data were normalized to the cell number.

### Immunoprecipitation (IP)
Total cell lysates were centrifuged at 13000 g for 15 min. at 4 °C. The supernatants were then incubated for 2 h with 4 μg of monoclonal ANTI-FLAG M2 antibody (Sigma-Aldrich) or control IgG under constant rotation at 4 °C. Immunoprecipitates were washed four times with the Lysis Buffer. Beads were finally collected by centrifugation at 5000 g for 2 min and brought up in of 20 μl of sample buffer for further analysis.

### Cardiac output assessment
Echocardiographic imaging was performed using the Vevo 3100 high-frequency ultrasound system (FUJIFILM VisualSonics, Inc.) equipped with the high-resolution MX400 linear array transducer. This transducer operates within a frequency range of 20–46 MHz and provides an axial resolution of 50μm. Standard cardiovascular views were obtained, including the parasternal long-axis (PLAX) and parasternal short-axis (SAX) views. All images were initially acquired in B-mode for anatomical orientation. Correct transducer positioning and visualization of cardiac structures were confirmed using color Doppler mode. M-mode images were used for measurements of cardiac output. M-mode imaging was performed in both PLAX and SAX views. Echocardiographic data were analyzed using Vevo LAB software, version 3.2.5 (FUJIFILM VisualSonics, Inc.).

### ATP, ADP and AMP quantification
The cell pellets or tumor samples were dissolved in 80 μL of 30% methanol in acetonitrile with 0.1 mM ammonium acetate and 0.01% NH4OH. As internal standard: Adenosin-13C10 5′-triphosphat (741167-1MG Sigma) Adenosine-15N5 5′-diphosphate (741167-1MG Sigma) Adenosin-13C10,15N5-5′-monophosphat (650676-1MG Sigma) were used. The mixture was snap-frozen and thawed three times in liquid nitrogen. The resulting mixture was centrifuged at $13,000 \times g$ for 10 min, and the supernatant was transferred to a glass vial. LC-MS/MS analysis was performed using liquid chromatography-tandem mass spectrometry on an ultra-performance liquid chromatography system (Aquity I-class, Waters) coupled to a triple quadrupole linear ion trap mass spectrometer (QTRAP 5500, Sciex). For normal phase chromatography, an AtlantisPremier BEH Z-HILIC 1.7 μm $2.1 \times 100$ mm Column from Waters was used. The mobile phase consisted of eluent A (95% acetonitrile, 10 mM ammonium acetate, and 0.01% NH4OH) and eluent B (40% acetonitrile, 10 mM ammonium acetate, and 0.01% NH4OH). Chromatographic separation was achieved at 40 °C with the following gradient program: Eluent B, from 0% to 100% within 13 min; 100% from 13 to 18 min; 0% from 18 to 25 min. The flow rate was set at 0.300 mL/min. The metabolites were analyzed in multiple reaction monitoring (MRM) scan mode using negative electrospray ionization (ESI). The ion source parameters were as follows: curtain gas (40 psi), ESI voltage (−4000V), source temperature (500 °C), gas 1 (70 psi), and gas 2 (60 psi). Compound-dependent source and fragmentation parameters were set to a 100 V declustering potential, 10 V entrance potential, 45 V collision energy, and 10 V cell exit potential. Data acquisition was performed using Analyst 1.7 (Sciex) and MultiQuant software, and data processing was done using the Sciex OS-MQ software package. Internal standards were used for quantification.

### Statistics and reproducibility
Each experiment was repeated independently at least twice -very often many more times- with similar results. For in-vivo experiments, analysis of 4 mice per group was performed. For immunohistochemistry analysis of patients samples: 14 nevi, 38 primary melanoma and 10 metastatic melanoma samples were used. For immunoblotting analysis of patients samples: lysates prepared from either tumor-adjacent normal (N) or malignant (M) tissues of five melanoma patients was used.

No statistical method was used to predetermine sample size. No data were excluded from the analyses. Animals were randomly assigned in treatment groups. The Investigators performing echocardiography, troponin T assay, ATP/ADP/AMP quantification were blinded to allocation during experiments and outcome assessment.

The Student's t test was used to test the significance of differences in different experimental conditions. GraphPad Prism 7 was used for statistical analysis and data plotting. $p < 0.05$ was set as a significance level.

### The Cancer Genome Atlas (TCGA) analyses
The correlations between *HK2*, *MCL1*, *BCL2*, and *BCL2L1* mRNA expression levels in human skin cutaneous melanoma (SKCM) were analyzed using the TIMER2.0 web platform (http://compbio.cn/timer2/), which integrates genomic data from The Cancer Genome Atlas (TCGA). A total of 471 melanoma samples were included in the analysis. The p-values and Spearman's correlation coefficients (Rho) were obtained directly from the TIMER2.0 portal, with Rho values representing the strength and direction of the correlations between gene expression levels.

### Reporting summary
Further information on research design is available in the Nature Portfolio Reporting Summary linked to this article.

## Data availability

All data generated in this study are included in the Source Data File or the Supplementary Information. Source data are provided with this paper.

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

## Acknowledgements

We are thankful to Dr. Manja Wobus for providing AML cells and to Dr. David Huang and Dr. Anna García-Saéz for providing WT and Bax/Bak DKO MEFs and to Dr Marco Herold for providing the hu*Mcl-1* mice. We are also thankful for Dr. Tomáš Mráček and Dr. Alena Pecinová for their kind help with metabolic analysis of heart tissues. M.E. has received funding from the European Research Council (ERC) under Horizon 2020 research and innovation programme (Onco-Energetics_OFF, grant agreement no. 852761) and the German Research Foundation (Project No. EL 1081/2-1) and has received partial support from Czech Science Foundation (GACR) (project no. 19-22156Y). L.S. has received funding from The Czech Science Foundation (GACR) Lead Agency (Weave) (Project No. 22-16819 K) and Czech-BioImaging large RI project (LM2023050 funded by MEYS CR) and GACR European Regional Development Fund (Project No. CZ.02.1.01/0.0/0.0/16_013/0001775) supported imaging lab and instrument funding. P. V.-F. has been partially supported as a student by Charles University grant SVV 260519/2023. L.M. and W.G. were partially supported by the National Institute for Cancer Research (Programme EXCELES, ID Project No. LX22NPO5102).

## Author contributions

M.E. and W.G. designed and conceived the study; W.G, B.D., E.H., C.S., V.A., F.F., M.S., K.G. F.B and D.E.S. performed and supervised biochemical and molecular biology experiments; P.P., M.D., J.H. P.V.F, J.K., J.P., P.M., M.E. and L.H.L contributed and oversaw mouse breeding and experiments, echocardiography, plasma isolation, FDG-PET imaging and in-vivo toxicity assays; G.B. and C.M. performed the immunohistochemistry of patient samples; S.T. and W.W. performed LC-MS/MS analysis; M.E., M.B., S.M., L.S. and L.M. contributed to funding acquisition, conceptual discussions and supervision of the work. The manuscript was written by M.E. with contributions from all authors. The final version of the manuscript was approved by all authors.

## Funding

## Competing interests

The authors declare no competing interests.

## Additional information

[1]Institute for Clinical Chemistry and Laboratory Medicine, University Hospital and Faculty of Medicine, Technische Universität Dresden, Dresden, Germany. [2]Medical Clinic I, University Hospital Carl Gustav Carus, Technische Universität Dresden, Dresden, Germany. [3]Mildred-Scheel Early Career Center, National Center for Tumor Diseases Dresden (NCT/UCC) University Hospital and Faculty of Medicine, Technische Universität Dresden, Dresden, Germany. [4]Cancer Cell Biology, Institute of Molecular Genetics of the Czech Academy of Sciences, Prague, Czech Republic. [5]Department of Cell Biology, Faculty of Science, Charles University, Prague, Czech Republic. [6]Center for Advanced Preclinical Imaging (CAPI), First Faculty of Medicine, Charles University, Prague, Czech Republic. [7]Institute of Pathological Physiology, First Faculty of Medicine, Charles University, Prague, Czech Republic. [8]Institute of Macromolecular Chemistry, Czech Academy of Sciences, Prague, Czech Republic. [9]Institute of Pharmacology and Toxicology, Medizinische Fakultät Carl Gustav Carus, Technische Universität Dresden, Dresden, Germany. [10]Department of Visceral, Thoracic and Vascular Surgery, University Hospital Carl Gustav Carus Dresden, Dresden, Germany. [11]National Center for Tumor Diseases (NCT), Partner Site Dresden, Dresden, German Cancer Research Center (DKFZ), Heidelberg, Germany. [12]Helmholtz-Zentrum Dresden—Rossendorf (HZDR), Dresden, Germany. [13]Department of Cardiology, University Hospital Heidelberg, Heidelberg, Germany. [14]German Center for Cardiovascular Research (DZHK), Heidelberg, Mannheim, Germany. [15]German Cancer Research Center (DKFZ), Heidelberg, Germany. [16]Department of Functional and Evolutionary Ecology, Molecular Systems Biology (MOSYS), University of Vienna, Vienna, Austria. [17]Vienna Metabolomics Center (VIME), University of Vienna, Vienna, Austria. [18]Section of Pathological Anatomy, Department of Medicine, Surgery and Neuroscience, University Hospital of Siena, Siena, Italy. [19]Department of Experimental Oncology, IEO European Institute of Oncology IRCCS, Milan, Italy; Department of Hemato-Oncology, Università Statale di Milano, Milan, Italy. ✉e-mail: mohamed.elgendy@ukdd.de

