## [Transparent Peer Review file · Nature Communications]

MCL1 Modulates mTORC1 Signaling to Promote Bioenergetics and Tumorigenesis

Corresponding Author: Dr Mohamed Elgendy

Version 0:

Reviewer comments:

Reviewer #1

(Remarks to the Author)

Manuscript Number:

Gui et al. identified mTORC1 is regulated by myeloid cell leukemia-1 (MCL1) in different cancer types using both in vitro and in vivo preclinical models. The authors identified MCL-1 as a potential regulator of mTORC1 using an in vitro phosphokinase screen. The authors used cultured melanoma cancer cell lines and human melanoma tissue samples to validate their screening data. First, knockdown or activation of MC1 in cultured cells was used to dissect the potential regulation of mTORC1 in vitro using protein quantification by western blotting and establish the consequences for mitochondrial respiration using Seahorse assays. Second, glucose uptake was quantified using xenograft mouse models to dissect the metabolic consequences in vivo. Combining preclinical models with translational data from human tissue samples and mechanistic dissections in vitro strengthens this study. However, significant shortcomings in the technical execution of experiments, the lack of controls, and limited in vivo validation dampen enthusiasm for the study. The following major concerns have been identified:

Major concerns:

1. The presentation and quantitation of western blotting and RT-qPCR results must be completed throughout the manuscript. None of the presented blots have protein loading or quantifications. It is recommended to include total protein stains for each blot to indicate the correct loading of samples and account for potential variability. Likewise, RT-qPCR data is missing appropriate normalization.
2. The 18FDG-PET imaging in shMCL1-treated xenograft models is valuable. However, there are several concerns regarding the data presentation and quality. The FDG-PET imaging in Figure 3I has no quantification, and a scale is missing. Normalization for injected FDG and blood glucose are missing. Furthermore, gating or normalization of FDG uptake in internal organs is missing (e.g., kidney, heart, brain). The authors need to provide CT images alongside the PET gating. PET imaging data needs to be quantified alongside gamma counting to demonstrate that both datasets confirm the increase in glucose uptake. Furthermore, gamma counting needs to be normalized by the dry weight of tumor samples.
3. Experimental evidence from in vitro cancer cells is inconsistently presented. For example, mitochondrial respiration in response to MCL1 knockdown is demonstrated in SK-Mel-30 melanoma cells, but lactate production and glucose consumption are quantified in A-375 and MOLM-13 cells (Figure 3). mRNA expression levels are measured in CHL-1 cells (Figure 4). The comparison of leukemia and melanoma cell lines is valuable, but the authors must consolidate their datasets and consistently compare them. Additional datasets are needed to validate the conclusions in Figure 3-5 using all four cell lines.
4. The authors provide in vitro evidence that MCL1 knockdown reduces glucose metabolism by inhibiting mTORC1 in AML and melanoma cell lines. These conclusions need to be supported with additional evidence quantifying glucose oxidation using tracer labeling or enzymatic assays). The authors provide some evidence using FDG-PET imaging supporting in vivo glucose uptake, but experiments were conducted inconsistently with different cell lines. Therefore, these datasets remain inconclusive.
5. Figure 3D shows an increased ADP to ATP ratio, suggesting that in SK-Mel-30 cells, knockdown of MCL1 is decreasing ATP provision. The authors must quantify the absolute ADP and ATP levels alongside AMP levels. Further, these findings need to be validated in vivo and expanded to include measurements of AMP. The increased ADP to ATP ratio potentially suggests an increased utilization of ATP rather than a reduced provision of nutrients. The current data does not support the conclusion that MCL1 knockdown reduces oxidative metabolism. Lastly, cell culture conditions can independently impact nutrient consumption. Mitochondrial respiration should be validated on at least one AML and melanoma cell line from in vivo tissue samples.

6. The authors indicate that inhibition or knockdown of MCL1 is associated with reduced activation of mTORC1, which promotes a metabolic switch and altered oxidative metabolism. mTORC1 is a known regulator of HK-2, which may explain the observed phenotypes mechanistically. However, a major limitation in the study design is the inconsistency of cell cultures and in vitro tests. Figures 3 and 5 demonstrate that the knockdown of MCL1 is cancer cell type dependent. shMCL1 does not impact mitochondrial respiration in CHL-1 cells, while SK-Mel-30 cells show reduced OCR. HK2 overexpression likely increases glucose uptake and utilization independent of MCL1.

7. The author tested the potential cardiotoxicity of MCL1 inhibitors and the possibility of rescuing the phenotype using leucine supplementation. These experiments are incomplete and need to be supported by an assessment of cardiac function to confirm the phenotype. Cardiac troponin T is a valuable biomarker but not conclusive without a functional reduction in cardiac output or other "clinical" signs. Furthermore, the impairment of mTORC1 activity and glucose metabolism via MCL1 inhibitors needs to be quantified and compared to the in vitro assessments in cancer studies. mTORC1 inhibition is known to promote physiologic hypertrophy in models of rapamycin inhibition or TSC2-KO, which is associated with increased glycolytic flux and glucose oxidation. The authors need to provide evidence that the loss of mTORC1 drives the potential cardiotoxicity in vivo and conduct metabolic studies.

Minor concerns:

1. The authors must homogenize their data presentation and include individual data points for bar graphs. Figure 7 depicts individual data sets, while Figures 3 and 4 do not.
2. The authors use bioenergetics inconsistently throughout the manuscript. For example, seahorse assays do not measure cellular 'bioenergetics' but mitochondrial respiration.

Reviewer #2

(Remarks to the Author)

In this study by Gui et al, a role for MCL1 in regulation of mTORC1 signalling to control metabolism is described. Convincing data are presented that support modulation of mTORC1 by MCL1 independently of apoptosis, and this seems to be mediated via TSC2 and Sestrin2 and is not common to other anti-apoptotic proteins BCL2 and BCL-XL. The authors also show that MCL1 mediated regulation of mTORC1 modulates metabolism with effects on oxidative phosphorylation and glycolysis demonstrated in vitro, and glycolysis in vivo. Notably, co-depletion of TSC2 rescued the effects on metabolism, however co-treatment with an apoptosis inhibitor did not, strongly supporting the notion that the effects of MCL1 on metabolism are downstream of its regulation of mTORC1 signalling and are independent of apoptosis.

The authors then further explore potential mechanisms of MCL1 mediated regulation of bioenergetics and show that downregulation of HK2 is required for MCL1 regulation of metabolism, and that HK2 overexpression is sufficient to partially reverse the tumour suppressing effects of MCL1 depletion in vivo.

Finally, the authors use these observations to test the idea that metabolic effects of mTOR may underpin cardiotoxicity that occurs after treatment with MCL1 inhibitors. By co-treating with leucine, they nicely show a rescue of cardiotoxicity induced by MCL1 inhibitors. Whilst this is a very interesting and potentially significant observation, it remains unclear the specific relationship between mTORC1 downregulation in this phenotype and whether leucine supplementation is specifically acting by reversing downregulation of mTORC1 signalling after inhibition of MCL1, or if it is functioning via a different mechanism.

In general the data presented support the conclusions, however in some cases additional evidence is required to fully support the novel aspects of the work (see specific comments below).

Overall, this is a nice piece of work that highlights a new mechanism underpinning apoptosis independent functions of MCL1 in metabolism and would be of sufficient interest to a broad readership. Notwithstanding these strengths, there are some weaknesses that should be addressed to ensure the robustness of the conclusions made in the study. Please see my specific comments below.

Figure 1: Loading controls are missing for western blots. If S6 levels are being used in place of a loading control, then this should be clearly stated as to why it's appropriate to use as a loading control (ie. Clearly showing that it doesn't change in these specific experimental settings, which would require an independent loading control not linked with signalling changes being assessed in the set of experiments). This should be clearly defined in the Figure Legend and/or methods otherwise readers cannot adequately interpret the data.

Figure 2: Same issue with loading controls; missing on multiple westerns however B-Actin and tubulin present for B and C? Its surprising that shTSC2 does not seem to affect mTORC1 signalling activity alone. Can the authors comment?

Figure 3 and S3: Ensuring consistency of data shown for different cell models across different experiments is warranted, and justification for use of different models for specific sets of experiments should be provided. This is particularly relevant given the different melanoma cell lines used in the study have different mutational contexts which should be considered. For example, different functional metabolism studies have been performed for different melanoma cell lines at different time points in the primary figure, whereby seahorse assays are performed for SKMEL-30 (BRAF wildtype, NRAS mutant) at 72hrs, whilst only lactate production is shown for A375 cells (BRAF mutant) at 24hrs. The SKMEL-30 cells are used for the in vivo experiment, and then the A375 cells are used for the TSC2 rescue studies in seahorse. Although seahorse

experiments have been performed for all 3 melanoma cells (Fig S3), it is confusing for the reader to keep track of what model has been used in the different experiments the way it is currently presented. It would be helpful if the different data panels in the figures were labelled with the cell line to make this clear to the reader. Justification of the use of different cell lines in different experimental contexts should also be provided given they all have different mutational backgrounds (A375 BRAF mutant; SKMEL-30 BRAF wildtype, NRAS mutant; CHL1 cells BRAF/NRAS wildtype, p53 mutant).

Figure 4: Can the authors explain why the follow up mechanism studies are performed in CHL1 cells (BRAF/NRAS wildtype, p53 mutant)? This model wasn't used for the in vivo or TSC2 rescue studies in Figure 3.

Is HK2 regulated by MCL1 in the other melanoma cell lines? These experiments are essential to ensure this mechanism is not a cell line specific effect in CHL1 cells.

Also, how is HK2 specifically regulated downstream of the MCL1-mTORC1 axis, as opposed to other key metabolic regulators tested?

Figure 5: The authors should consider showing the data and statistics in Fig S5C-F in the primary figure, as it is difficult to interpret what is going on from the raw seahorse profiles currently shown in Fig 5A, especially since the OCR profiles behave strangely after FCCP injection. The representative profiles would be better placed in Fig S5 instead.

Figure 7: Whilst these data are interesting it remains unclear whether leucine supplementation is specifically acting by reversing downregulation of mTORC1 signalling after inhibition of MCL1, or if it is functioning via a different mechanism. Loading control also appears to be missing for the western blot.

General comment: The authors should consider changing the title of the manuscript given there is no evidence of any signalling activity underpinning regulation of mTORC1 by MCL1 and this could be misleading. Rather than MCL1 signals to mTORC1 it could simply state "MCL1 modulates or regulates".

Reviewer #3

(Remarks to the Author)

The manuscript by Elgendy et al reports crosstalk between the anti-apoptotic regulator MCL1 and the mTORC1 signaling pathway. There are some very nice features of the paper, but some outstanding issues that must be addressed.

Strengths:

1. The use of both cell lines, primary patient samples, and multiple cancer cell types to illustrate the repression of mTORC1 signaling induced by shMCL1 treatment is robust.
2. The use of caspase inhibitors to repress cell death in Figure 2 helps to convince that the effects of shMCL1 are not due to subliminal cell death induction. However, the long incubation (72 hours) may lead to some uninhibited caspase activation (see weakness #2 below).
3. The epistasis experiments that establish that Sestrin2 and TSC2 are necessary to see repression of mTORC1 upon shMCL1 silencing are well performed.

Weaknesses:

1. The choice of MCL1 inhibitors is absolutely unacceptable. While 8-10 years ago these inhibitors were best in class, there are now much more potent and on target inhibitors that are in or have been in clinical trials. The on-target specificity of these new inhibitors is much clearer and these are the ones that MUST be used. These include S68345, AMG176, AZD5991, ABBV-467, and others. Thus, all experiments in Figures 6 and 7 should be replaced with state-of-the-art inhibitors to make this finding impactful.
2. To further enhance the concept that the loss of MCL1 does not induce subliminal cell death that leads to changes in cellular signaling prior to overt signs of apoptosis, the authors should consider using BAX and BAK doubly-deficient cells. These cells fail to activate mitochondrial outer membrane permeabilization even after long-term culture with apoptotic inducers.
3. While the authors have done a good job of genetically ordering that loss of MCL1 represses mTORC1 in a Sestrin2 and TSC2 mediated manner, it is still unclear how this occurs mechanistically. The fact that the authors contend that pharmacological inhibitors (see concerns in weakness #1 above) induce similar changes would suggest that the pathway depends on the MCL1's hydrophobic BH3-binding pocket, which is the target of all MCL1 inhibitors. If this is the case, then the authors would have to contend that something (perhaps Sestrin2) is binding to this pocket. No such mechanistic studies are provided.
4. MCL1 inhibitors are well known to have lower affinity for murine MCL1 than human MCL1, making their use in mouse models challenging (see weakness #1 above). Thus, the Troponin release experiments in Figure 7 should be carefully considered and single doses are not acceptable. It is unclear whether the doses used would be considered efficacious. The authors should monitor WBC numbers as reported by Kotschy et al. Nature (2016) as a pharmacodynamic marker.

Version 1:

Reviewer comments:

Reviewer #1

(Remarks to the Author)

The authors have made an effort to address the reviewers concerns and significantly improved the manuscript. Additional controls and experiments were included in this revision, which address several of the major concerns. Furthermore, the authors made an effort to improve the presentation of their datasets. It is further appreciated that the authors indicate limitations of the study in their discussion and clearly indicate when they could not reproduce findings. The following concerns should be addressed in a revised manuscript:

Major Concerns

1. A previous review raised concerns regarding the lack of evidence for MCL1 knockdown to reduce glucose metabolism by inhibiting mTORC1. It was suggested to quantify glucose oxidation using either enzymatic assays or tracer studies. The authors conducted additional measurements and presenting in Supplementary Figure 3E and F glucose consumption and lactate production. The manuscript correctly describes these datasets, but it should be noted that these measurements do not quantify glucose oxidation as indicated in the rebuttal. Nonetheless, the data supports the authors conclusions and adds value to demonstrate reduced glucose metabolism. It is critical to represent the data as absolute values (e.g mmol/mL or mmol/mg protein) rather than fold changes. Otherwise it is difficult to compare between groups and assess the quality of the measurements.
2. 3. Figure 3H and Figure 3K depict ATP levels as fold changes. These data should be presented as absolute quantification to allow comparisons between cell types.
3. The authors measured cardiac function using echocardiographic imaging. These measurements need to be included in the supplementary data and main figures. The cardiac output is valuable, but critical measures of cardiac function are missing. Please include, ejection fraction, fractional shortening, LVIDd, LVIDs, LVEDV, LVESV, Lvmass, LV Vol d, and LV s. The ejection fraction, fractional shortening and Cardiac output can be presented as graphs while the remaining dataset can be included as a table.

Minor Concerns

1. The authors state in the introduction “Deregulating energetics and evading apoptosis are two hallmarks of cancer that need to be closely coordinated.” This sentence is misleading because the term “energetics” is outdated. Perhaps the author meant the following:
“Deregulated energy substrate metabolism and evading apoptosis are two hallmarks of cancer that need to be closely coordinated.”
2. Graphs in Supp. Figure 3 depicting Seahorse measurements require revision. The greyscale color palette is too pale making it difficult to identify shBcl-2 and shBcl-XL readings.
3. Supp. Figure 3 Panel R: y-axis is partially covering x-axis labels.

Reviewer #2

(Remarks to the Author)

The authors have adequately addressed all my concerns, and the data presented in the revised manuscript is significantly more robust and strongly supports the conclusions made. Consolidation of cell line models across different experimental contexts, and reference to their distinct mutational background, allows better interpretation of the findings. All missing controls are now included.

I have no further comments.

Reviewer #3

(Remarks to the Author)

The authors did an excellent job in responding with new experimental data to my prior concerns. They should be commended for an excellent study that helps define yet another amazing feature of MCL1 biology.

Point-by-point reply to reviewers' comments

(Our responses are highlighted in Red)

Reviewer #1 (Remarks to the Author); expert in cardiotoxicity and metabolism:

Gui et al. identified mTORC1 is regulated by myeloid cell leukemia-1 (MCL1) in different cancer types using both in vitro and in vivo preclinical models. The authors identified MCL-1 as a potential regulator of mTORC1 using an in vitro phosphokinase screen. The authors used cultured melanoma cancer cell lines and human melanoma tissue samples to validate their screening data. First, knockdown or activation of MCL1 in cultured cells was used to dissect the potential regulation of mTORC1 in vitro using protein quantification by western blotting and establish the consequences for mitochondrial respiration using Seahorse assays. Second, glucose uptake was quantified using xenograft mouse models to dissect the metabolic consequences in vivo. Combining preclinical models with translational data from human tissue samples and mechanistic dissections in vitro strengthens this study. However, significant shortcomings in the technical execution of experiments, the lack of controls, and limited in vivo validation dampen enthusiasm for the study. The following major concerns have been identified:

We thank the reviewer for their comments. Over the last year, we have conducted a series of experiments to thoroughly address the concerns they raised and a substantial amount of new data has been added in the revised manuscript, which we believe significantly improved the quality of the study.

Major concerns:

1. The presentation and quantitation of western blotting and RT-qPCR results must be completed throughout the manuscript. None of the presented blots have protein loading or quantifications. It is recommended to include total protein stains for each blot to indicate the correct loading of samples and account for potential variability. Likewise, RT-qPCR data is missing appropriate normalization.

We have now included loading controls using housekeeping proteins in addition to the previously included total levels of S6 and S6K (that are often used as loading controls in reference to mTORC1 activity). We have also quantified the blots and fold changes are presented under the blots in the figures. RT-qPCR data was normalized to the housekeeping control beta actin and presented as fold changes relative to the control (scr). We have now clearly indicated that in the figure legend.

2. The ¹⁸F-FDG-PET imaging in shMCL1-treated xenograft models is valuable. However, there are several concerns regarding the data presentation and quality. The FDG-PET imaging in Figure 3I has no quantification, and a scale is missing. Normalization for injected FDG and blood glucose are missing.

Furthermore, gating or normalization of FDG uptake in internal organs is missing (e.g., kidney, heart, brain). The authors need to provide CT images alongside the PET gating. PET imaging data needs to be quantified alongside gamma counting to demonstrate that both datasets confirm the increase in glucose uptake. Furthermore, gamma counting needs to be normalized by the dry weight of tumor samples.

We thank the reviewer for these important suggestions, which we believed significantly improved the ¹⁸FDG-PET imaging analysis. We have strictly followed the suggestions and implemented changes accordingly. In addition to gamma counting normalized to the dry weight of tumors (which we believe is a very accurate method to precisely quantify ¹⁸FDG uptake), we now show quantified Maximum Standardized Uptake Values (SUV_{max}) in Supp. Fig. 3N, O and Fig. 3L, M. The two analyses show consistent results. The data shown is normalized to injected ¹⁸FDG. Blood glucose levels are now shown and quantified ¹⁸FDG uptake in internal organs (kidney, heart, brain and liver) have been added (Supp. Figure 3 P, Q). Furthermore, and as discussed in the Results section, establishing the control and MCL1-depleted tumors on both flanks of mice controlled for the inter-mouse and inter-organ variabilities of ¹⁸FDG uptake as it allowed comparison of two tumors established in the same mouse. We now present the data as lines connecting the values of control and MCL1-depleted tumors established in the same mice for better comparison.

3. Experimental evidence from in vitro cancer cells is inconsistently presented. For example, mitochondrial respiration in response to MCL1 knockdown is demonstrated in SK-Mel-30 melanoma cells, but lactate production and glucose consumption are quantified in A-375 and MOLM-13 cells (Figure 3). mRNA expression levels are measured in CHL-1 cells (Figure 4). The comparison of leukemia and melanoma cell lines is valuable, but the authors must consolidate their datasets and consistently compare them. Additional datasets are needed to validate the conclusions in Figure 3-5 using all four cell lines.

We thank the reviewer for this important point. While our aim from using different cell systems was to show the generalizability of our findings, we agree that this may have confused the message. Therefore, we have now focused our study on melanoma as a study model and reproduced all datasets in two melanoma cell lines CHL-1 and SK-MEL30, taking into consideration the scope and feasibility of experimental work.

4. The authors provide in vitro evidence that MCL1 knockdown reduces glucose metabolism by inhibiting mTORC1 in AML and melanoma cell lines. These conclusions need to be supported with additional evidence quantifying glucose oxidation (using tracer labeling or enzymatic assays). The authors provide some evidence using FDG-PET imaging supporting in vivo glucose uptake, but experiments were conducted inconsistently with different cell lines. Therefore, these datasets remain inconclusive.

We have now used enzymatic assay to quantify glucose oxidation as suggested by the reviewer. As shown in Supp. Figure 3E and F, MCL1 knockdown reduced glucose metabolism in both of the study models CHL-1 and SK-MEL30 cells. This *in-vitro* data supports the *in-vivo* FDG-PET data of tumors derived from CHL-1 and SK-MEL30 cells.

5. Figure 3D shows an increased ADP to ATP ratio, suggesting that in SK-Mel-30 cells, knockdown of MCL1 is decreasing ATP provision. The authors must quantify the absolute ADP and ATP levels alongside AMP levels. Further, these findings need to be validated in vivo and expanded to include measurements of AMP. The increased ADP to ATP ratio potentially suggests an increased utilization of ATP rather than a reduced provision of nutrients. The current data does not support the conclusion that MCL1 knockdown reduces oxidative metabolism. Lastly, cell culture conditions can independently impact nutrient

consumption. Mitochondrial respiration should be validated on at least one AML and melanoma cell line from *in vivo* tissue samples.

We have now quantified absolute ADP, ATP and AMP levels in CHL-1 and SK-MEL30 tumors and cells using LC-MS/MS analysis as requested by the reviewer. Our data show that depletion of MCL1 both *in vitro* and *in vivo* resulted in decrease in ATP levels without significant increase in ADP or AMP levels, suggesting reduction in ATP production (Figure 3H, K; Supp. Figure 3H; Supp. Figure 3K-M).

We totally agree with the reviewer about the importance of *in vivo* validation metabolic data generated *in vitro*. To this end, we have assessed mitochondrial respiration of tumors derived from CHL-1 or SK-MEL30 cells and consistent with the *in vitro* results, MCL1 depletion resulted in dramatic inhibition of mitochondrial respiration of tumors *in vivo* (Figure 3J and Supp. Figure 3J).

6. The authors indicate that inhibition or knockdown of MCL1 is associated with reduced activation of mTORC1, which promotes a metabolic switch and altered oxidative metabolism. mTORC1 is a known regulator of HK-2, which may explain the observed phenotypes mechanistically. However, a major limitation in the study design is the inconsistency of cell cultures and *in vitro* tests. Figures 3 and 5 demonstrate that the knockdown of MCL1 is cancer cell type dependent. shMCL1 does not impact mitochondrial respiration in CHL-1 cells, while SK-Mel-30 cells show reduced OCR. HK2 overexpression likely increases glucose uptake and utilization independent of MCL1.

As mentioned in point 3, we have now reproduced all datasets in CHL-1 and SK-MEL30 cells. In both models, MCL-1 knockdown reduced *HK2* expression in mTORC1-dependent manner (Figure 4 A, B and Supp. Figure 4A). Additionally, in three other melanoma cell lines, MCL1 knockdown led to reduction in HK2 levels (Supp. Figure 4C), suggesting that it is not cell line specific.

Quantification of mitochondrial respiration from at least 3 repeats show that knockdown of MCL1 in CHL-1 and SK-MEL30 cells and tumors consistently resulted in significant inhibition of mitochondrial respiration (Figure 3A-C; Supp Figure 3 A,B; Figure 3J and Supp. Figure 3J).

We agree with the reviewer on the crucial role of HK2 in glucose metabolism. Our data show specific modulation of *HK2* expression by MCL1-mTORC1 axis in both CHL-1 and SK-MEL 30 cell models (Figure 4A and Supp. Figure 4A). This effect was specific to MCL1 as it was not shared with not Bcl-2 or Bcl-xL (Figure 4C and Supp. Figure 4B) and seemed to be specific to HK2 as among the array of metabolic regulators we screened, it was the only one modulated by MCL1 in mTORC1-depedent manner. To investigate whether the modulation of HK2 by MCL1-mTORC1 axis contributes to the modulation of the observed effect on cellular metabolism, we aimed to rescue HK2 downregulation by ectopic overexpression and examined whether this would impact the inhibitory effect of MCL1 Knockdown on metabolism and importantly on cell viability and tumor growth. Taking into account the suggestion of the reviewer, we now tested the overexpression of two other glycolysis regulators PKM2 and ALDOA. Our data shows that overexpression of PKM2 and ALDOA failed to reproduce the same effect of HK2 overexpression on metabolism and viability in MCL1-depleted cells (Figure 5A, B, D; Supp. Figure 5A-C), pointing out for a more crucial role for HK2 modulation in this context. As mentioned now in the Discussion section, our data however does not exclude the possibility that additional mechanisms of regulation or crosstalk may exist between MCL1-

mTORC1 and metabolic pathways especially that HK2 overexpression only partially rescued the inhibitory effect of MCL1 knockdown on mitochondrial respiration and glycolysis.

7. The author tested the potential cardiotoxicity of MCL1 inhibitors and the possibility of rescuing the phenotype using leucine supplementation. These experiments are incomplete and need to be supported by an assessment of cardiac function to confirm the phenotype. Cardiac troponin T is a valuable biomarker but not conclusive without a functional reduction in cardiac output or other “clinical” signs. Furthermore, the impairment of mTORC1 activity and glucose metabolism via MCL1 inhibitors needs to be quantified and compared to the in vitro assessments in cancer studies. mTORC1 inhibition is known to promote physiologic hypertrophy in models of rapamycin inhibition or TSC2-KO, which is associated with increased glycolytic flux and glucose oxidation. The authors need to provide evidence that the loss of mTORC1 drives the potential cardiotoxicity in vivo and conduct metabolic studies.

We thank the reviewer for raising these important points. In the revised manuscript we now include data generated in humanized *Mcl-1* mice, in which murine *Mcl-1* was replaced by its human homolog. This circumvented the limitation of low affinity of newly-introduced MCL1 inhibitors against murine *Mcl-1* and enabled the precise prediction of metabolic consequences of MCL1 inhibition in our study.

We have now assessed cardiac output as a clinical sign of cardiac function as suggested by the reviewer. The changes in cardiac output were consistent with the observed changes in cardiac Troponin T (Figure 7A; Supp. Figure 7C; Supp. Figure 8C, D).

We have also quantified the impairment of mTORC1 in the hearts of mice treated with MCL1 inhibitors (Figure 7C; Supp. Figure 8E). Our data points out to a comparable inactivation of mTORC1 to that observed in cancer studies (Figure 6A-F).

We agree with the reviewer that mTORC1 signaling plays crucial roles in the physiology and pathology of the heart. Importantly, we now show that rapamycin treatment almost completely reversed the rescue of mTORC1 signaling by leucine and subsequently reversed the cardio-protective effect of leucine in mice treated with MCL1 inhibitors (Figure 7A-C ; Supp. Figure 7C ; Supp. Figure 8 C-E). We have also conducted metabolic studies analyzing mitochondrial respiration in the heart tissues from all groups of mice. We now show significant impairment of mitochondrial respiration in the hearts of humanized *Mcl-1* mice upon MCL1 inhibition, an effect that was rescued by leucine and rapamycin reversed the rescue effect of leucine (Figure 7B), all consistent and tightly correlating with the observed modulations on mTORC1 signaling and cardiac function.

We agree with the reviewer that the role of mTORC1 is rather complex. Conflicting reports implicated roles for mTORC1 activation and inhibition in mediating cardiotoxicity. We now discussed these reports in more details in the Discussion part. As the reviewer pointed out, mTORC1 activation via inducible cardiac-specific TSC2 knockdown results in physiologic hypertrophy. However, while mTORC1 evidently becomes activated upon TSC2 ablation, other mTOR-independent effects of TSC2 inactivation have been reported. Probably the most direct link between mTOR signaling and cardiotoxicity is the observations that cardiac specific, kinase-dead mTOR transgenic mice showed significantly reduced cardiac function and that mice with deficient myocardial mTORC1 activity, by targeted ablation of mTORC1 essential

component raptor, suffered from deteriorated cardiac functions. The final contribution of mTORC1 to the physiology and pathology of unstressed and stressed hearts is likely to be context-dependent. Differences in the specific models used, experimental conditions, treatment duration and dosage may all play crucial roles.

Importantly, in our model of mice treated with MCL1 inhibitors-/Leucine +/- Rapamycin, we observed a tight correlation between the magnitude of mTORC1 inactivation and increases in Troponin T and decrease in cardiac output strongly suggesting that mTORC1 inactivation contributes to the cardiotoxicity of MCL1 inhibitors.

Minor concerns:

1. The authors must homogenize their data presentation and include individual data points for bar graphs. Figure 7 depicts individual data sets, while Figures 3 and 4 do not.

We totally agree with the reviewer and indeed thank them for this suggestion. We have now homogenized all data presentation and included data points for all bar graphs.

2. The authors use bioenergetics inconsistently throughout the manuscript. For example, seahorse assays do not measure cellular 'bioenergetics' but mitochondrial respiration.

Since the focus of the study is on bioenergetics, we made use of the Real-Time ATP production rate assay of the Seahorse, an assay that enables the simultaneous quantification of real-time ATP production from glycolysis and mitochondria. We have also quantified the absolute levels of ATP, AMP and ADP as suggested by the reviewer and amended the text to be more specific on what exactly is assessed in each experiment.

Reviewer #2 (Remarks to the Author); expert in melanoma and metabolism:

In this study by Gui et al, a role for MCL1 in regulation of mTORC1 signalling to control metabolism is described. Convincing data are presented that support modulation of mTORC1 by MCL1 independently of apoptosis, and this seems to be mediated via TSC2 and Sestrin2 and is not common to other anti-apoptotic proteins BCL2 and BCL-XL. The authors also show that MCL1 mediated regulation of mTORC1 modulates metabolism with effects on oxidative phosphorylation and glycolysis demonstrated in vitro, and glycolysis in vivo. Notably, co-depletion of TSC2 rescued the effects on metabolism, however co-treatment with an apoptosis inhibitor did not, strongly supporting the notion that the effects of MCL1 on metabolism are downstream of its regulation of mTORC1 signalling and are independent of apoptosis.

The authors then further explore potential mechanisms of MCL1 mediated regulation of bioenergetics and show that downregulation of HK2 is required for MCL1 regulation of metabolism, and that HK2 overexpression is sufficient to partially reverse the tumour suppressing effects of MCL1 depletion in vivo.

Finally, the authors use these observations to test the idea that metabolic effects of mTOR may underpin cardiotoxicity that occurs after treatment with MCL1 inhibitors. By co-treating with leucine, they nicely

show a rescue of cardiotoxicity induced by MCL1 inhibitors. Whilst this is a very interesting and potentially significant observation, it remains unclear the specific relationship between mTORC1 downregulation in this phenotype and whether leucine supplementation is specifically acting by reversing downregulation of mTORC1 signalling after inhibition of MCL1, or if it is functioning via a different mechanism.

In general the data presented support the conclusions, however in some cases additional evidence is required to fully support the novel aspects of the work (see specific comments below).

Overall, this is a nice piece of work that highlights a new mechanism underpinning apoptosis independent functions of MCL1 in metabolism and would be of sufficient interest to a broad readership. Notwithstanding these strengths, there are some weaknesses that should be addressed to ensure the robustness of the conclusions made in the study.

We thank the reviewer for their comments, the changes implemented based on their comments greatly improved the manuscript.

Figure 1: Loading controls are missing for western blots. If S6 levels are being used in place of a loading control, then this should be clearly stated as to why it's appropriate to use as a loading control (ie. Clearly showing that it doesn't change in these specific experimental settings, which would require an independent loading control not linked with signalling changes being assessed in the set of experiments). This should be clearly defined in the Figure Legend and/or methods otherwise readers cannot adequately interpret the data.

Figure 2: Same issue with loading controls; missing on multiple westerns however B-Actin and tubulin present for B and C?

We have now included loading controls using housekeeping proteins.

It's surprising that shTSC2 does not seem to affect mTORC1 signalling activity alone. Can the authors comment?

We have now quantified all blots. Quantification of 3 independent repeats shows around 2-3 fold increase in pS6 and pS6k levels upon TSC2 knockdown relative to control (scr) (Figure 2E).

Of note, mTORC1 is known to be hyperactive in melanoma cells and TSC2 knockdown may thus have less dramatic effect on further activation of mTORC1 as compared to other cell types with lower basal mTORC1 activity.

Figure 3 and S3: Ensuring consistency of data shown for different cell models across different experiments is warranted, and justification for use of different models for specific sets of experiments should be provided. This is particularly relevant given the different melanoma cell lines used in the study have different mutational contexts which should be considered. For example, different functional metabolism studies have been performed for different melanoma cell lines at different time points in the primary

figure, whereby seahorse assays are performed for SKMEL-30 (BRAF wildtype, NRAS mutant) at 72hrs, whilst only lactate production is shown for A375 cells (BRAF mutant) at 24hrs. The SKMEL-30 cells are used for the in vivo experiment, and then the A375 cells are used for the TSC2 rescue studies in seahorse. Although seahorse experiments have been performed for all 3 melanoma cells (Fig S3), it is confusing for the reader to keep track of what model has been used in the different experiments the way it is currently presented. It would be helpful if the different data panels in the figures were labelled with the cell line to make this clear to the reader. Justification of the use of different cell lines in different experimental contexts should also be provided given they all have different mutational backgrounds (A375 BRAF mutant; SKMEL-30 BRAF wildtype, NRAS mutant; CHL1 cells BRAF/NRAS wildtype, p53 mutant).

We thank the reviewer for this important point. While our aim from using different cell models was to demonstrate that the phenotype is not confined to a certain cell model, we agree that this may have confused the readers. Therefore, we have now focused our study on melanoma as a study model and reproduced all datasets in two melanoma models CHL-1 and SK-MEL30 cells.

In figures where more cell models are used, we have labeled the figure panels with the name of cells used.

Figure 4: Can the authors explain why the follow up mechanism studies are performed in CHL1 cells (BRAF/NRAS wildtype, p53 mutant)? This model wasn't used for the in vivo or TSC2 rescue studies in Figure 3.

As mentioned above, we have now consistently reproduced all datasets in CHL-1 and SK-MEL30 cells.

Is HK2 regulated by MCL1 in the other melanoma cell lines? These experiments are essential to ensure this mechanism is not a cell line specific effect in CHL1 cells.

We now demonstrate that MCL1 knockdown consistently results in reduction in HK2 levels in five melanoma cell lines: CHL-1, SK-MEL-30, MeWo, IGR1 and IPC298 (Figure 4B,C; Supp. Figure 4B,C) indicating that it is not cell line-specific.

Also, how is HK2 specifically regulated downstream of the MCL1-mTORC1 axis, as opposed to other key metabolic regulators tested?

The limited-scale screening we have done identified *HK2* to be specifically modulated by MCL1-mTORC1 axis (Figure 4). Our analysis showed that HK2 is regulated at the transcriptional level. Indeed, HK2 is an established target of mTOR signaling as has long been shown by numerous studies.

Roberts DJ, Miyamoto S. Hexokinase II integrates energy metabolism and cellular protection: Akt-ing on mitochondria and TORCing to autophagy. *Cell Death Differ.* 2015 Feb;22(2):248-57. doi: 10.1038/cdd.2014.173. Epub 2014 Oct 17. Erratum in: *Cell Death Differ.* 2015 Feb;22(2):364. doi: 10.1038/cdd.2014.208. PMID: 25323588; PMCID: PMC4291497.

Bhaskar PT, Nogueira V, Patra KC, Jeon SM, Park Y, Robey RB, Hay N. mTORC1 hyperactivity inhibits serum deprivation-induced apoptosis via increased hexokinase II and GLUT1 expression, sustained Mcl-1 expression, and glycogen synthase kinase 3beta inhibition. *Mol Cell Biol.* 2009 Sep;29(18):5136-47. doi: 10.1128/MCB.01946-08. Epub 2009 Jul 20. PMID: 19620286; PMCID: PMC2738274.

Vogt C, Ardehali H, Iozzo P, Yki-Järvinen H, Koval J, Maezono K, Pendergrass M, Printz R, Granner D, DeFronzo R, Mandarino L. Regulation of hexokinase II expression in human skeletal muscle in vivo. *Metabolism*. 2000 Jun;49(6):814-8. doi: 10.1053/meta.2000.6245. PMID: 10877213.

Duarte AI, Santos P, Oliveira CR, Santos MS, Rego AC. Insulin neuroprotection against oxidative stress is mediated by Akt and GSK-3beta signaling pathways and changes in protein expression. *Biochim Biophys Acta*. 2008 Jun;1783(6):994-1002. doi: 10.1016/j.bbamcr.2008.02.016. Epub 2008 Feb 29. PMID: 18348871.

Osawa H, Sutherland C, Robey RB, Printz RL, Granner DK. Analysis of the signaling pathway involved in the regulation of hexokinase II gene transcription by insulin. *J Biol Chem*. 1996 Jul 12;271(28):16690-4. doi: 10.1074/jbc.271.28.16690. PMID: 8663315.

Lee AW, States DJ. Colony-stimulating factor-1 requires PI3-kinase-mediated metabolism for proliferation and survival in myeloid cells. *Cell Death Differ*. 2006 Nov;13(11):1900-14. doi: 10.1038/sj.cdd.4401884. Epub 2006 Mar 3. PMID: 16514418.

Culbert AA, Tavaré JM. Multiple signalling pathways mediate insulin-stimulated gene expression in 3T3-L1 adipocytes. *Biochim Biophys Acta*. 2002 Oct 11;1578(1-3):43-50. doi: 10.1016/s0167-4781(02)00481-5. PMID: 12393186.

Chehtane M, Khaled AR. Interleukin-7 mediates glucose utilization in lymphocytes through transcriptional regulation of the hexokinase II gene. *Am J Physiol Cell Physiol*. 2010 Jun;298(6):C1560-71. doi: 10.1152/ajpcell.00506.2009. Epub 2010 Mar 3. PMID: 20200205; PMCID: PMC2889638.

Figure 5: The authors should consider showing the data and statistics in Fig S5C-F in the primary figure, as it is difficult to interpret what is going on from the raw seahorse profiles currently shown in Fig 5A, especially since the OCR profiles behave strangely after FCCP injection. The representative profiles would be better placed in Fig S5 instead.

We have now reproduced the Seahorse profiles and as suggested by the reviewer, we now present the seahorse profiles side by side with bar graphs of basal and maximal respiration for easier interpretation.

Figure 7: Whilst these data are interesting it remains unclear whether leucine supplementation is specifically acting by reversing downregulation of mTORC1 signalling after inhibition of MCL1, or if it is functioning via a different mechanism. Loading control also appears to be missing for the western blot.

We agree with the reviewer that this is a crucial point. As discussed in the Discussion section, mTORC1 signaling plays crucial roles in the physiology and pathology of the heart. Importantly, we now show that rapamycin treatment almost completely reversed the rescue of mTORC1 signaling by leucine and subsequently reversed the cardio-protective effect of leucine in mice treated with MCL1 inhibitors (Figure 7A-C ; Supp. Figure 7C ; Supp. Figure 8 C-E).

We have also quantified the impairment of mTORC1 in the hearts of mice in different groups (Figure 7C; Supp. Figure 8E). In mice treated with MCL1 inhibitors-/Leucine +/- Rapamycin, we observed a tight correlation between the magnitude of mTORC1 inactivation and increases in Troponin T and decrease in cardiac output.

General comment: The authors should consider changing the title of the manuscript given there is no evidence of any signalling activity underpinning regulation of mTORC1 by MCL1 and this could be misleading. Rather than MCL1 signals to mTORC1 it could simply state "MCL1 modulates or regulates".

We thank the reviewer for this suggestion. We totally agree and we have changed the title as per their suggestion.

Reviewer #3 (Remarks to the Author); expert in Mcl1:

The manuscript by Elgendy et al reports crosstalk between the anti-apoptotic regulator MCL1 and the mTORC1 signaling pathway. There are some very nice features of the paper, but some outstanding issues that must be addressed.

Strengths:

1. The use of both cell lines, primary patient samples, and multiple cancer cell types to illustrate the repression of mTORC1 signaling induced by shMCL1 treatment is robust.

We thank the reviewer for highlighting the strengths of the study and their encouraging comments.

2. The use of caspase inhibitors to repress cell death in Figure 2 helps to convince that the effects of shMCL1 are not due to subliminal cell death induction. However, the long incubation (72 hours) may lead to some uninhibited caspase activation (see weakness #2 below).

While 72 hours were needed to achieve efficient knockdown, caspase inhibitor was added only in the last 16 hours prior to analysis, as we now clarify in the figure legends.

3. The epistasis experiments that establish that Sestrin2 and TSC2 are necessary to see repression of mTORC1 upon shMCL1 silencing are well performed.

Weaknesses:

1. The choice of MCL1 inhibitors is absolutely unacceptable. While 8-10 years ago these inhibitors were best in class, there are now much more potent and on target inhibitors that are in or have been in clinical trials. The on-target specificity of these new inhibitors is much clearer and these are the ones that MUST be used. These include S68345, AMG176, AZD5991, ABBV-467, and others. Thus, all experiments in Figures 6 and 7 should be replaced with state-of-the-art inhibitors to make this finding impactful.

We thank the reviewer for this suggestion. We agree and we have now replaced the figures using ABBV-467 suggested by the reviewer. Additionally, we included the data produced with UMI-77 in supplementary figures. UMI-77 is a newer specific inhibitor of Mcl-1 that has been used in a number of recent studies. (e.g. Cen et al Nat Commun. 2020; Thomalla et al., Blood. 2022; Post et al., Haematologica 2022). It is also commercially available at affordable cost, which we believe will allow more groups to reproduce our data and follow up on our study.

2. To further enhance the concept that the loss of MCL1 does not induce subliminal cell death that leads to changes in cellular signaling prior to overt signs of apoptosis, the authors should consider using BAX and BAK doubly-deficient cells. These cells fail to activate mitochondrial outer membrane permeabilization even after long-term culture with apoptotic inducers.

We have carried out this experiment. We now show that MCL1 knockdown results in similar inactivation of mTORC1 in BAX and BAK doubly-deficient cells as compared to their WT counterparts (Supp. Figure 2B). This adds to the array of evidence showing that mTORC1 regulation by MCL1 is independent of apoptosis.

3. While the authors have done a good job of genetically ordering that loss of MCL1 represses mTORC1 in a Sestrin2 and TSC2 mediated manner, it is still unclear how this occurs mechanistically. The fact that the authors contend that pharmacological inhibitors (see concerns in weakness #1 above) induce similar changes would suggest that the pathway depends on the MCL1's hydrophobic BH3-binding pocket, which is the target of all MCL1 inhibitors. If this is the case, then the authors would have to contend that something (perhaps Sestrin2) is binding to this pocket. No such mechanistic studies are provided.

We thank the reviewer for this very interesting suggestion. Indeed, our data indeed shows that MCL1 binds Sestrin 2 and that MCL1's hydrophobic BH3-binding pocket is required for this binding (Supp. 6 A,B). While the main message of the study is to unravel novel role for MCL1 in the modulation of mTORC1 signaling, we now provide additional mechanistic insights on the role of Sestrin 2 in mediating this effect. Further mechanistic analysis on the interplay between MCL1 and Sestrin 2 will be an interesting topic for future studies.

4. MCL1 inhibitors are well known to have lower affinity for murine MCL1 than human MCL1, making their use in mouse models challenging (see weakness #1 above). Thus, the Troponin release experiments in Figure 7 should be carefully considered and single doses are not acceptable. It is unclear whether the doses used would be considered efficacious. The authors should monitor WBC numbers as reported by Kotschy et al. Nature (2016) as a pharmacodynamic marker.

We agree with the reviewer and to circumvent the limitation of low affinity of MCL1 inhibitors against murine *Mcl-1*, we have used humanized *Mcl-1* mice, in which murine *Mcl-1* was replaced by its human homolog, which enabled the precise prediction of metabolic consequences of MCL1 inhibition in our study. We have used two doses of MCL1 inhibitors as requested by the reviewer and we have assessed the effects of these doses on growth of tumor xenografts (Supp. Figure 7A; Supp. Figure 8A), mTORC1 signaling in tumor xenografts (Supp. Figure 7B; Supp. Figure 8B) and in heart tissues (Figure 7C and Supp. Figure 8E), cardiac functions (Figure 7A; Supp. Figure 7C), respiration of heart tissues (Figure 7B) and as suggested by the reviewer on WBC as well as on RBC and PLT (Supp. Figure 7D-F).

Reviewer #1 (Remarks to the Author):

We thank the reviewer for their constructive comments that ultimately improved our study.

Major Concerns

1. A previous review raised concerns regarding the lack of evidence for MCL1 knockdown to reduce glucose metabolism by inhibiting mTORC1. It was suggested to quantify glucose oxidation using either enzymatic assays or tracer studies. The authors conducted additional measurements and presenting in Supplementary Figure 3E and F glucose consumption and lactate production. The manuscript correctly describes these datasets, but it should be noted that these measurements do not quantify glucose oxidation as indicated in the rebuttal. Nonetheless, the data supports the authors conclusions and adds value to demonstrate reduced glucose metabolism. It is critical to represent the data as absolute values (e.g mmol/mL or mmol/mg protein) rather than fold changes. Otherwise it is difficult to compare between groups and assess the quality of the measurements.

We have now replaced the fold changes data and included the absolute values in the figures. The results remain consistent and the values are what to be expected from our previous studies and reports in literature, ensuring the robustness of measurements.

2. 3. Figure 3H and Figure 3K depict ATP levels as fold changes. These data should be presented as absolute quantification to allow comparisons between cell types.
3. The authors measured cardiac function using echocardiographic imaging. These measurements need to be included in the supplementary data and main figures. The cardiac output is valuable, but critical measures of cardiac function are missing. Please include, ejection fraction, fractional shortening, LVIDd, LVIDs, LVEDV, LVESV, Lvmass, LV Vol d, and LV s. The ejection fraction, fractional shortening and Cardiac output can be presented as graphs while the remaining dataset can be included as a table.

We have now included in a table format the measures obtained from echocardiography (Supp. Fig. 8c and Supp. Tables 1 and 3), in addition to the cardiac output graph previously requested by the reviewer that was already included in the manuscript.

The echocardiography measures are certainly valuable and are consistent with our findings and support the Troponin T data. Furthermore, we would like to mention that in the particular context of MCL1 inhibitors, Troponin T data are of the most direct clinical relevance as the increase in Troponin T values was the reason of the FDA hold on clinical testing of MCL1

inhibitors.

Minor Concerns

1. The authors state in the introduction “Deregulating energetics and evading apoptosis are two hallmarks of cancer that need to be closely coordinated.” This sentence is misleading because the term “energetics” is outdated. Perhaps the author meant the following: “Deregulated energy substrate metabolism and evading apoptosis are two hallmarks of cancer that need to be closely coordinated.”

We changed the text as suggested.

2. Graphs in Supp. Figure 3 depicting Seahorse measurements require revision. The greyscale color palette is too pale making it difficult to identify shBcl-2 and shBcl-XL readings.

We replaced the graph with a colored one.

3. Supp. Figure 3 Panel R: y-axis is partially covering x-axis labels.

We thank the reviewer for their keen eye on details and for spotting this mistake which we now corrected.

Reviewer #2 (Remarks to the Author):

The authors have adequately addressed all my concerns, and the data presented in the revised manuscript is significantly more robust and strongly supports the conclusions made. Consolidation of cell line models across different experimental contexts, and reference to their distinct mutational background, allows better interpretation of the findings. All missing controls are now included. I have no further comments.

We thank the reviewer for their constructive comments and we are happy that we managed to address their important remarks.

Reviewer #3 (Remarks to the Author):

The authors did an excellent job in responding with new experimental data to my prior concerns. They should be commended for an excellent study that helps define yet another amazing feature of MCL1 biology.

We are deeply pleased by the reviewer's very kind commandment. THANK YOU!